# Quantum Speedups for Minimax Optimization and Beyond

**Chengchang Liu**
The Chinese University of Hong Kong
7liuchengchang@gmail.com

**Zongqi Wan**[*]
Great Bay University
zqwan@gbu.edu.cn

**Jialin Zhang**
Institute of Computing Technology, CAS
zhangjialin@ict.ac.cn

**Xiaoming Sun**
Institute of Computing Technology, CAS
sunxiaoming@ict.ac.cn

**John C.S. Lui**
The Chinese University of Hong Kong
cslui@cse.cuhk.edu.hk

## Abstract

This paper investigates convex-concave minimax optimization problems where only the function value access is allowed. We introduce a class of Hessian-aware quantum zeroth-order methods that can find the $\epsilon$-saddle point within $\tilde{\mathcal{O}}(d^{2/3}\epsilon^{-2/3})$ function value oracle calls. This represents an improvement of $d^{1/3}\epsilon^{-1/3}$ over the $\mathcal{O}(d\epsilon^{-1})$ upper bound of classical zeroth-order methods, where $d$ denotes the problem dimension. We extend these results to $\mu$-strongly-convex $\mu$-strongly-concave minimax problems using a restart strategy, and show a speedup of $d^{1/3}\mu^{-1/3}$ compared to classical zeroth-order methods. The acceleration achieved by our methods stems from the construction of efficient quantum estimators for the Hessian and the subsequent design of efficient Hessian-aware algorithms. In addition, we apply such ideas to non-convex optimization, leading to a reduction in the query complexity compared to classical methods.

## 1 Introduction

We consider the following unconstrained minimax problem:

$$\min_{\mathbf{x} \in \mathbb{R}^m} \max_{\mathbf{y} \in \mathbb{R}^n} h(\mathbf{x}, \mathbf{y}), \tag{1}$$

where $h(\cdot, \cdot)$ is convex in $\mathbf{x}$ and concave in $\mathbf{y}$. Let $d = m + n$, $\mathbf{z} = (\mathbf{x}^\top, \mathbf{y}^\top)^\top \in \mathbb{R}^d$, and define $f(\mathbf{z}) \triangleq h(\mathbf{x}, \mathbf{y})$. The above problem has received considerable attention in the field of machine learning due to its wide applications, including fairness-aware learning [52, 32], AUC maximization [51, 20], robust optimization [3], game theory [48], and reinforcement learning [15, 43].

There are numerous classical algorithms to solve the convex-concave minimax problem (1). First-order methods, such as the optimistic gradient descent ascent (OGDA) [44, 36], extra gradient method (EG) [27, 38], along with their variants [40, 39] find an $\epsilon$-saddle point within $\mathcal{O}(\epsilon^{-1})$ gradient oracle queries when $h(\cdot, \cdot)$ is smooth. When $\nabla^2 h(\cdot, \cdot)$ is Lipschitz continuous, second-order

---

[*]The corresponding author.

39th Conference on Neural Information Processing Systems (NeurIPS 2025).

methods offer faster convergence rates. The Newton proximal extra gradient method (NPE) [37] and its cubic-regularized realization [31, 22] require $\mathcal{O}(\epsilon^{-2/3})$ queries to the second-order oracle. Very recently, Chen et al. [11] further improved the second-order oracle complexity to $\tilde{\mathcal{O}}(\epsilon^{-4/7})$. These results have been generalized to the cases where the $p$-th order derivative of $h(\cdot, \cdot)$ is Lipschitz, with query complexity of $\mathcal{O}(\epsilon^{-\frac{p}{p+1}})$ to the $p$-th order oracle [30, 23, 24].

However, access to the gradient oracle or higher-order oracles of $h(\cdot, \cdot)$ is not always available, as the calculation of the exact gradient (or higher order) information can be expensive or even infeasible [42, 35, 25, 53]. This necessitates the design of efficient derivative-free algorithms to solve equation (1). Beznosikov et al. [4] proposed a gradient-free algorithm with $\mathcal{O}(d\epsilon^{-2})$ query complexity to $h(\cdot, \cdot)$ for the smooth convex-concave minimax problem (1). Subsequently, this rate was improved to $\mathcal{O}(d\epsilon^{-1})$ by Sadiev et al. [45], whose dependence on $\epsilon^{-1}$ matches the lower bound for first-order methods [54].

The aforementioned methods for minimax problems are all designed for classical computing machines. However, the advantage of quantum computing has been investigated for various optimization problems. Accessing quantum counterparts of the classical oracles always leads to better query complexities or even a breakthrough over the classical lower bounds, including convex [47, 8, 9, 55] or non-convex optimization [46, 19], semi-definite programming [5, 6], non-smooth optimization [8, 33, 28], stochastic optimization [41, 55, 46], and so on. There are also some previous studies on some specific minimax problems, including the zero-sum game [29, 16] and minimizing the maximal loss [49]. However, it remains an open question whether quantum speedup is available for general convex-concave minimax problems. Thus, the first question we aim to address is:

*Can quantum zeroth-order methods be designed to surpass the $\mathcal{O}(d\epsilon^{-1})$ query complexity of classical zeroth-order methods, thereby demonstrating a quantum speedup for general convex-concave minimax problems?*

A natural approach to achieve this is to leverage the fast gradient estimator [26], which can approximate the gradient of a smooth objective function within $\tilde{\mathcal{O}}(1)$ queries to the quantum function value oracle, based on the quantum Fourier transform [50]. In convex optimization, employing the fast gradient estimator within cutting-plane algorithms and gradient descent leads to improved query complexities compared to classical zeroth-order methods. Specifically, the quantum cutting-plane method [8] and the quantum gradient descent method [2] achieve query complexities of $\tilde{\mathcal{O}}(d)$ and $\tilde{\mathcal{O}}(\epsilon^{-1})$, respectively, improving upon the $\tilde{\mathcal{O}}(d^2)$ and $\tilde{\mathcal{O}}(d\epsilon^{-1})$ complexities of classical zeroth-order methods by a factor of $d$. For non-convex optimization, a similar strategy has been used in the design of quantum perturbed AGD [19], resulting in an improved query complexity of $\tilde{\mathcal{O}}(\epsilon^{-7/4})$ compared to the $\tilde{\mathcal{O}}(d\epsilon^{-7/4})$ complexity of the classical zeroth-order method [53]. These methods use the quantum gradient estimators to achieve the same query complexities as classical first-order methods by accessing only zeroth-order oracles, which reduces the dependency on $d$. On the other hand, the oracle complexity of classical second-order methods enjoys a better dependency on $\epsilon^{-1}$ when compared to the classical first-order methods. This motivates us to ask:

*Can we go beyond quantum estimation for the gradient and design efficient Hessian-aware quantum zeroth-order algorithms with better dependence on both $\epsilon^{-1}$ and $d$?*

In this paper, we provide an affirmative answer to the above two questions. To this end, we develop a quantum estimator for the Hessian matrix using the finite difference method and design a novel Hessian-aware zeroth-order optimization framework. We summarize our contributions as follows.

- For the convex-concave problem, we propose a Hessian-aware quantum zeroth-order method (HAQZO), with query complexity of $\tilde{\mathcal{O}}(d\epsilon^{-2/3})$ to find the $\epsilon$-saddle point, which surpasses the classical zeroth-order method by a factor of $\epsilon^{-1/3}$. We further accelerate such query complexity to $\tilde{\mathcal{O}}(d + d^{2/3}\epsilon^{-2/3})$ by proposing a double-loop Hessian-aware quantum method (HAQZO$^+$) that can reuse the Hessian estimators. HAQZO$^+$ accelerates the classical algorithms in terms of $\epsilon^{-1}$ and $d$. We compare HAQZO and HAQZO$^+$ with the existing method in Table 1. The detailed analysis of HAQZO and HAQZO$^+$ can be found in Sections 4.1 and 4.2, respectively.

- For the strongly-convex-strongly-concave problem, we apply the restart strategy in HAQZO$^+$ and propose Restart-HAQZO$^+$. We prove that Restart-HAQZO$^+$ finds the $\epsilon$-point with query complexity of $\tilde{\mathcal{O}}(d + d^{2/3}(L_2/\mu)^{2/3})$, outperforms the classical method by a factor $d^{1/3}\mu^{-1/3}$.

Table 1: We summarize the complexities of function value oracles to find the $\epsilon$-saddle point (c.f. Section 2) for the convex-concave minimax problem (1).

| Methods | Oracle | Query Complexity | Reference |
|---|---|---|---|
| ZOSPA | classical | $\mathcal{O}\left(d\epsilon^{-2}\right)$ | Beznosikov et al. [4] |
| ZOVIA | classical | $\mathcal{O}\left(d\epsilon^{-1}\right)$ | Sadiev et al. [45] |
| HAQZO Algorithm 3 | quantum | $\tilde{\mathcal{O}}\left(d\epsilon^{-2/3}\right)$ | Theorem 4.3 |
| HAQZO$^+$ Algorithm 4 | quantum | $\tilde{\mathcal{O}}\left(d + d^{2/3}\epsilon^{-2/3}\right)$ | Theorem 4.5 |

Table 2: We summarize the complexities of function value oracles to find the $\epsilon$-point for $\mu$-strongly-convex-$\mu$-strongly-concave minimax problem (1), i.e. $\|\mathbf{z} - \mathbf{z}^*\|^2 \leq \epsilon$. We use $L_i$ ($i = 1, 2$) denotes the Lipschitz continuous parameter of $i$-th order derivatives of $f(\cdot)$.

| Methods | Oracle | Query Complexity | Reference |
|---|---|---|---|
| ZOVIA | classical | $\tilde{\mathcal{O}}\left(dL_1/\mu\right)$ | Sadiev et al. [45] |
| Restart-HAQZO$^+$ Algorithm 5 | quantum | $\tilde{\mathcal{O}}\left(d + d^{2/3}(L_2/\mu)^{2/3}\right)$ | Theorem 4.8 |

Table 3: We summarize the complexities of function value oracles to find the $\epsilon$-stationary point of non-convex minimization problem (10), i.e. $\|\nabla f(\mathbf{z})\| \leq \epsilon$. We use $d$ to denote the dimension of the problem.

| Methods | Oracle | Query Complexity | Reference |
|---|---|---|---|
| GFM | classical | $\mathcal{O}\left(d\epsilon^{-7/4}\right)$ | Zhang and Gu [53] |
| DF-CNM | classical | $\tilde{\mathcal{O}}\left(d^2\epsilon^{-3/2}\right)$ | Cartis et al. [7] |
| Zero-Order CNM | classical | $\tilde{\mathcal{O}}\left(d^2 + d^{3/2}\epsilon^{-3/2}\right)$ | Doikov and Grapiglia [13] |
| Q-Perturbed-AGD | quantum | $\tilde{\mathcal{O}}\left(\epsilon^{-7/4}\right)$ | Gong et al. [19] |
| QCNM Algorithm 6 | quantum | $\tilde{\mathcal{O}}\left(d + d^{1/2}\epsilon^{-3/2}\right)$ | Theorem 5.2 |

The comparison of the query complexities can be found in Table 2 and the detailed analysis is presented in Section 4.3.

- We further generalize the design of Hessian-aware quantum methods to solve non-convex problems with Lipschitz continuous Hessian. We propose the quantum cubic regularized-Newton method (QCNM) with query complexity of $\tilde{\mathcal{O}}(d + d^{1/2}\epsilon^{-3/2})$ to find the $\epsilon$-stationary point, which is better than all classical zeroth-order algorithms. The proposed QCNM method also enjoys an improved quantum query complexity over the existing state-of-the-art quantum algorithm when $d = \mathcal{O}(\epsilon^{-1/2})$, demonstrating the power of designing Hessian-aware quantum algorithms. We compare QCNM with existing classical algorithms and quantum algorithms in Table 3 and present the results in Section 5.

## 2 Preliminaries

We make the following assumptions on $f(\mathbf{z}) \triangleq h(\mathbf{x}, \mathbf{y})$.

**Assumption 2.1.** We assume $f(\mathbf{z}) = h(\mathbf{x}, \mathbf{y})$ is convex in $\mathbf{x}$ and concave in $\mathbf{y}$.

**Assumption 2.2.** We assume the $f(\cdot)$, $\nabla f(\cdot)$, $\nabla^2 f(\cdot)$ are $L_0$, $L_1$, and $L_2$-Lipschitz continuous, respectively, i.e. we have $|f(\mathbf{z}) - f(\mathbf{z}')| \leq L_0 \|\mathbf{z} - \mathbf{z}'\|$, $\|\nabla f(\mathbf{z}) - \nabla f(\mathbf{z}')\| \leq L_1 \|\mathbf{z} - \mathbf{z}'\|$, and

$$\left\| \nabla^2 f(\mathbf{z}) - \nabla^2 f(\mathbf{z}') \right\| \leq L_2 \|\mathbf{z} - \mathbf{z}'\|, \tag{2}$$

for any $\mathbf{z}, \mathbf{z}' \in \mathbb{R}^d$.

We aim to find the approximate saddle point [39, 31, 12], which is defined as follows.

**Definition 1** (Nesterov [39]). *Let $\mathbb{B}_\beta(\mathbf{w})$ be the ball centered at $\mathbf{w}$ with radius $\beta$. Let $\mathbf{z}^* \triangleq \begin{bmatrix} \mathbf{x}^* \\ \mathbf{y}^* \end{bmatrix}$ be the saddle point of function $f(\cdot)$. For a given point $\hat{\mathbf{z}} \triangleq \begin{bmatrix} \hat{\mathbf{x}} \\ \hat{\mathbf{y}} \end{bmatrix}$, we let $\beta$ sufficiently large such that $\max\{\|\hat{\mathbf{x}} - \mathbf{x}^*\|, \|\hat{\mathbf{y}} - \mathbf{y}^*\|\} \leq \beta$ holds, we define the restricted gap function as*

$$\mathrm{Gap}(\hat{\mathbf{z}}; \beta) := \max_{\mathbf{y} \in \mathbb{B}_\beta(\mathbf{y}^*)} f\left( \begin{bmatrix} \hat{\mathbf{x}} \\ \mathbf{y} \end{bmatrix} \right) - \min_{\mathbf{x} \in \mathbb{B}_\beta(\mathbf{x}^*)} f\left( \begin{bmatrix} \mathbf{x} \\ \hat{\mathbf{y}} \end{bmatrix} \right),$$

*We call $\hat{\mathbf{z}}$ an $\epsilon$-saddle point if $\mathrm{Gap}(\hat{\mathbf{z}}; \beta) \leq \epsilon$ and $\beta = \Omega(\|\mathbf{z}_0 - \mathbf{z}^*\|)$.*

In the following context, we define $\mathbf{F}(\cdot)$ as $\mathbf{F}(\mathbf{z}) \triangleq \mathbf{J} \nabla f(\mathbf{z}) = \begin{bmatrix} \nabla_\mathbf{x} h(\mathbf{x}, \mathbf{y}) \\ -\nabla_\mathbf{y} h(\mathbf{x}, \mathbf{y}) \end{bmatrix}$, where $\mathbf{J} = \mathrm{diag}(\mathbf{I}_m, -\mathbf{I}_n)$. The Jacobian of $\mathbf{F}(\cdot)$ can be written as $\nabla \mathbf{F}(\mathbf{x}) = \mathbf{J} \nabla^2 \mathbf{F}(\mathbf{x})$. The following proposition shows that $\mathbf{F}(\cdot)$ is monotone if $f(\cdot)$ satisfies Assumption 2.1.

**Proposition 2.3** (Lemma 2.7 [30]). *If $f$ satisfies Assumption 2.1, then for all $\mathbf{z}, \mathbf{z}' \in \mathbb{R}^d$ it holds that $\langle \mathbf{F}(\mathbf{z}) - \mathbf{F}(\mathbf{z}'), \mathbf{z} - \mathbf{z}' \rangle \geq \mathbf{0}$. For a given $\hat{\mathbf{z}} \in \mathbb{R}^d$, its gap can be bounded by $\mathrm{Gap}(\hat{\mathbf{z}}; \beta) \leq \max_{\mathbf{z} \in \mathbb{B}_{\sqrt{2}\beta}(\mathbf{z}^*)} \langle \mathbf{F}(\hat{\mathbf{z}}), \hat{\mathbf{z}} - \mathbf{z} \rangle$.*

We define the quantum evaluation oracle for a function $f(\cdot)$.

**Definition 2.4** (Quantum Function Evaluation Oracle). A quantum evaluation oracle for a function $f$ is defined as the following unitary transformation

$$\mathbf{U}_f : |\mathbf{z}\rangle |\mathbf{v}\rangle \mapsto |\mathbf{z}\rangle |\mathbf{v} \oplus f(\mathbf{z})\rangle. \tag{3}$$

Here $\oplus$ is the bit-wise XOR operation. We say that we have a quantum evaluation oracle for $f$ with accuracy $\epsilon_0$ if we have a quantum evaluation oracle for $\tilde{f}$, such that $|f(\mathbf{z}) - \tilde{f}(\mathbf{z})| \leq \epsilon_0$ for all $\mathbf{z}$.

*Remark* 2.5. The quantum advantages are stated in terms of query complexity on the function evaluation oracle. In many situations, query complexity dominates the computational complexity of the algorithm, which is a natural setting in both classical and quantum optimization. For example, considering the generalized linear model such that $f(\mathbf{x}) = h(\mathbf{A}^\mathrm{T} \mathbf{x})$ where $\mathbf{A} \in \mathbb{R}^{d \times n}$. The circuit implementation of the oracle of $f$ may involve dominating computational complexity if $n \gg d$ in this example, and our algorithm achieves meaningful quantum speedups under such a setting.

## 3 Gradient and Hessian Estimation via Quantum Function Evaluation Oracle

Before introducing our quantum algorithms, we first introduce the quantum estimators for the gradient and Hessian of the objective function by using the quantum evaluation oracle on $f(\cdot)$, which are the critical components of our methods. These results are natural and direct extensions of Jordan's method [26] for the smooth objective. We do not consider them to be our primary technical contribution, but state them for completeness.

### 3.1 Quantum Gradient Estimator

Quantum gradient estimator is first proposed in [26] for the smooth objective, and its rigorous statement is given by Gilyén et al. [18], Chakrabarti et al. [8], van Apeldoorn et al. [47]. The following is one of the statements.

**Lemma 3.1** (Lemma 2.2 [8]). *Let $f$ be an $L_0$-Lipschitz continuous and $L_1$-smooth function. Given the access to a quantum evaluation oracle of $f$ with $\epsilon_0$ accuracy, then for $\epsilon_{\mathbf{g}} \geq \epsilon_0$ there is a quantum algorithm $\mathcal{A}(f, \epsilon_{\mathbf{g}}, L_0, L_1, \mathbf{z})$ which outputs an estimate $\widetilde{\nabla} f(\mathbf{z})$ of $\nabla f(\mathbf{z})$, satisfying that $\forall i \in [d]$, $\Pr\left( \left| \left[ \widetilde{\nabla} f(\mathbf{z}) \right]_i - [\nabla f(\mathbf{z})]_i \right| \geq 1500\sqrt{L_1 d \epsilon_{\mathbf{g}}} \right) \leq \frac{1}{3}$. Moreover, the $\mathcal{A}$ algorithm uses $\mathcal{O}(1)$ queries to the quantum evaluation oracle and $\mathcal{O}\left( d \log \frac{L_0}{dL_1\epsilon_{\mathbf{g}}} \right)$ quantum gates.*

The following lemma allows for an arbitrarily small failure probability $\delta \in (0, 1)$ to the quantum gradient estimator, which generalizes the results above.

**Lemma 3.2** (Quantum Gradient Estimator). *Let $f(\cdot)$ be a $L_0$-Lipschitz continuous and $L_1$-smooth function. Given the access to a quantum evaluation oracle of $f$ with $\epsilon_0$ accuracy, then for $\epsilon_{\mathbf{g}} \geq \epsilon_0$, there exists a quantum algorithm $\texttt{QuantumGradient}(f, \epsilon_{\mathbf{g}}, L_0, L_1, \mathbf{z}, \delta)$ which outputs an estimate $\widetilde{\nabla} f(\mathbf{z})$ of $\nabla f(\mathbf{z})$, satisfying*

$$\Pr\left( \left\| \widetilde{\nabla} f(\mathbf{z}) - \nabla f(\mathbf{z}) \right\|_2 \geq 1500 d\sqrt{L_1 \epsilon_{\mathbf{g}}} \right) \leq \delta. \tag{4}$$

*Moreover, $\texttt{QuantumGradient}$ uses $\mathcal{O}(\log(\frac{d}{\delta}))$ queries to $\mathbf{U}_f$ and $\mathcal{O}\left( d \log(\frac{L_0}{dL_1\epsilon_{\mathbf{g}}}) \log(\frac{d}{\delta}) \right)$ gates.*

## 3.2 Quantum Hessian-vector Estimator and Quantum Hessian Estimator

In this section, we show that the Hessian vector product of a smooth object function can also be constructed within the $\tilde{\mathcal{O}}(1)$ quantum function evaluation oracle. Furthermore, since $\nabla^2 f(\mathbf{z}) = [\nabla^2 f(\mathbf{z})\mathbf{e}_1, \cdots, \nabla^2 f(\mathbf{z})\mathbf{e}_d]$, the Hessian of a smooth object function can be constructed within the $\tilde{\mathcal{O}}(d)$ quantum function evaluation oracle.

We formally present our construction of the quantum Hessian vector product estimator in Algorithm 1 and state its complexity in the following lemma.

---

**Algorithm 1** $\texttt{QuantumHessianVector}(f, \epsilon_{\mathbf{hv}}, L_0, L_1, L_2, \mathbf{z}, \mathbf{v}, \delta)$

---

1: $M = \|\mathbf{v}\|_2$
2: $\Delta = 20\sqrt{15}\epsilon_{\mathbf{hv}}^{1/4} M^{-1/2} L_2^{-1/2} L_1^{1/4} d^{1/2}$
3: $\widetilde{\nabla} f(\mathbf{z}) := \texttt{QuantumGradient}(f, \epsilon_{\mathbf{hv}}, L_0, L_1, \mathbf{z}, \delta/2)$
4: $\widetilde{\nabla} f(\mathbf{z} + \Delta \mathbf{v}) := \texttt{QuantumGradient}(f, \epsilon_{\mathbf{hv}}, L_0, L_1, \mathbf{z} + \Delta \cdot \mathbf{v}, \delta/2)$
5: **Return** $\frac{1}{\Delta} \left( \widetilde{\nabla} f(\mathbf{x} + \Delta \mathbf{v}) - \widetilde{\nabla} f(\mathbf{x}) \right)$

---

**Lemma 3.3** (Quantum Hessian Vector Estimator). *Suppose $f$ satisfies Assumption 2.2. Given the access to $\mathbf{U}_f$ with $\epsilon_0$ accuracy, let $\mathbf{hv} = \texttt{QuantumHessianVector}(f, \epsilon_{\mathbf{hv}}, L_0, L_1, L_2, \mathbf{z}, \mathbf{v}, \delta)$ be the output of Algorithm 1 where $\epsilon_{\mathbf{hv}} \geq \epsilon_0$, then it holds that*

$$\Pr\left( \left\| \mathbf{hv} - \nabla^2 f(\mathbf{z})\mathbf{v} \right\|_2 > 10\sqrt{15}(dL_2 M)^{1/2}(\epsilon_{\mathbf{hv}} L_1)^{1/4} \right) \leq \delta.$$

*Moreover, Algorithm 1 uses $\mathcal{O}(\log(\frac{d}{\delta}))$ queries to $\mathbf{U}_f$ and $\mathcal{O}\left( d \log(\frac{L_0}{dL_1\epsilon_{\mathbf{hv}}}) \log(\frac{d}{\delta}) \right)$ gates.*

*Remark* 3.4. He et al. [21] proposed a quantum estimator for a row of a Hessian, which can be viewed as a special case of our quantum Hessian vector estimator. Besides, they do not provide results on the query complexity and gate complexity.

Given the Hessian vector estimator, we are ready to construct the Hessian estimator by calculating the estimators of the Hessian vector set $\{\nabla^2 f(\mathbf{x})\mathbf{e}_i\}_{i \in [d]}$, which is formally given in Algorithm 2. The following results show how well the output of Algorithm 2 approximates $\nabla^2 f(\mathbf{z})$.

**Lemma 3.5** (Quantum Hessian Estimator). *Suppose $f$ satisfies Assumption 2.2. Given access to a quantum evaluation oracle of $f$ with $\epsilon_0$ accuracy and $\epsilon_{\mathbf{H}} \geq \epsilon_0$, let $\widetilde{\nabla}^2 f(\mathbf{z}) = \texttt{QuantumHessian}(f, \epsilon_{\mathbf{H}}, L_0, L_1, L_2, \mathbf{z}, \delta)$ be the output of Algorithm 2, then it holds that*

$$\Pr\left( \left\| \widetilde{\nabla}^2 f(\mathbf{z}) - \nabla^2 f(\mathbf{z}) \right\|_2 > 10\sqrt{15}d^2 L_1^{1/4} L_2^{1/2} \epsilon_{\mathbf{H}}^{1/4} \right) \leq \delta. \tag{5}$$

*Moreover, Algorithm 2 uses $\mathcal{O}(d \log(\frac{d}{\delta}))$ queries to $\mathbf{U}_f$ and $\mathcal{O}\left( d^2 \log(\frac{L_0}{dL_1\epsilon_{\mathbf{H}}}) \log(\frac{d}{\delta}) \right)$ gates.*

---

**Algorithm 2** QuantumHessian($f, \epsilon_{\mathbf{H}}, L_0, L_1, L_2, \mathbf{z}, \delta$)

---

1: $\mathbf{H} = \mathbf{0}_{d \times d}$
2: **for** $i \in [d]$
3:    $\mathbf{H}[i,:] = $ QuantumHessianVector($f, \epsilon_{\mathbf{H}}, L_0, L_1, L_2, \mathbf{z}, \mathbf{e}_i, \delta/d$)
4: $\tilde{\mathbf{H}} = \frac{1}{2}(\mathbf{H} + \mathbf{H}^\top)$
5: **Return** $\tilde{\mathbf{H}}$

---

**Algorithm 3** HAQZO($\mathbf{z}_0, T, L_0, L_1, L_2, \delta$)

---

1: **for** $t = 0, \cdots, T-1$ **do**
2:    Choose $\epsilon_{1,t} > 0$ and $\epsilon_{\mathbf{H},t} > 0$
3:    $\tilde{\mathbf{g}}_t = $ QuantumGradient($f, \epsilon_{1,t}, L_0, L_1, \mathbf{z}_t, \delta/(3T)$)   and   $\mathbf{g}_t = \mathbf{J}\tilde{\mathbf{g}}_t$
4:    $\tilde{\mathbf{H}}_t = $ QuantumHessian($f, \epsilon_{\mathbf{H},t}, L_0, L_1, L_2, \mathbf{z}_t, \delta/(3T)$)   and   $\mathbf{H}_t = \mathbf{J}\tilde{\mathbf{H}}_t$
5:    Compute the inexact cubic step *i.e.* find $\mathbf{z}_{t+1/2}$ that satisfies

$$\mathbf{g}_t + \left(\mathbf{H}_t + 6(L_2\|\mathbf{z}_t - \mathbf{z}_{t+1/2}\| + \sqrt{1500}d^{1/2}L_1^{1/4}\epsilon_{1,t}^{1/4} + \sqrt{1500}d^2 L_1^{1/4}L_2^{1/2}\epsilon_{\mathbf{H},t}^{1/4})\mathbf{I}\right)(\mathbf{z}_{t+1/2} - \mathbf{z}_t) = \mathbf{0}$$

6:    $\lambda_t = 6\left(L_2\|\mathbf{z}_t - \mathbf{z}_{t+1/2}\| + \sqrt{1500}d^{1/2}L_1^{1/4}\epsilon_{1,t}^{1/4} + \sqrt{1500}d^2 L_1^{1/4}L_2^{1/2}\epsilon_{\mathbf{H},t}^{1/4}\right)$.
7:    Choose $\epsilon_{2,t} > 0$
8:    $\tilde{\mathbf{v}}_{t+1/2} = $ QuantumGradient($f, \epsilon_{2,t}, L_0, L_1, \mathbf{z}_{t+1/2}, \delta/(3T)$)   and   $\mathbf{v}_t = \mathbf{J}\tilde{\mathbf{v}}_t$
9:    $\mathbf{z}_{t+1} = \mathbf{z}_t - \lambda_t^{-1}\mathbf{v}_t$.
10: **end for**
11: **return** $\bar{\mathbf{z}}_T = \frac{1}{\sum_{t=0}^{T-1}\lambda_t^{-1}} \sum_{t=0}^{T-1} \lambda_t^{-1}\mathbf{z}_{t+1/2}$.

---

*Remark* 3.6. We note an independent work by Zhang and Shao [56], who also employed the finite difference method to construct a Hessian estimator for the more general class of complex analytical functions. In contrast to our estimator, which is designed for smooth real functions, their approach utilizes the more sophisticated spectral method to handle the complex case. On the other hand, our theoretical error bound is measured using the spectral norm ($\|\cdot\|_2$), while the bound in [56] is given in the infinity norm ($\|\cdot\|_\infty$).

## 4    Quantum Speedups for Minimax Optimization

In this section, we introduce quantum algorithms to find the $\epsilon$-saddle point for general convex-concave minimax problems. In Section 4.1, we propose a Hessian-aware algorithm with $\tilde{\mathcal{O}}(d\epsilon^{-2/3})$ queries to the quantum function evaluation oracle, which outperforms the classical state-of-the-art algorithm by a factor of $\epsilon^{-1/3}$. We further improve such query complexity to $\tilde{\mathcal{O}}(d^{2/3}\epsilon^{-2/3})$, which outperforms the classical algorithm by a factor of $d^{1/3}\epsilon^{-1/3}$, by proposing a double-loop algorithm that reuses the Hessian estimators in Section 4.2. In Section 4.3, we generalize our results to strongly-convex-strongly-concave problems.

### 4.1    Hessian-Aware Quantum Algorithm with Better Dependency on $\epsilon^{-1}$

Our idea is to use the quantum gradient estimator and the quantum Hessian estimator to obtain a close approximation of $\mathbf{F}(\mathbf{z})$ and $\nabla\mathbf{F}(\mathbf{z})$ and then apply the Newton proximal extragradient framework [37]. We present our Hessian-aware quantum zeroth-order method (HAQZO) in Algorithm 3.

To analyze Algorithm 3, we first consider the following generalized NPE update:

$$\begin{cases} \mathbf{z}_{t+1/2} = \mathbf{z}_t - (\mathbf{H}_t + \lambda_t\mathbf{I})^{-1}\mathbf{g}_t \\ \mathbf{z}_{t+1} = \mathbf{z}_t - \lambda_t^{-1}\mathbf{v}_t \end{cases}, \tag{6}$$

where $\mathbf{g}_t, \mathbf{v}_t$, and $\mathbf{H}_t$ are some approximations to $\mathbf{F}(\mathbf{z}_t), \mathbf{F}(\mathbf{z}_{t+1/2})$, and $\nabla\mathbf{F}(\mathbf{z}_t)$, which satisfy

$$\|\mathbf{g}_t - \mathbf{F}(\mathbf{z}_t)\| \leq \delta_{1,t}, \quad \|\mathbf{v}_t - \mathbf{F}(\mathbf{z}_{t+1/2})\| \leq \delta_{2,t}, \quad \text{and} \quad \|\mathbf{H}_t - \nabla\mathbf{F}(\mathbf{z}_t)\| \leq \delta_{\mathbf{H},t}. \tag{7}$$

The following lemma shows that the update of (6) recovers the convergence rates of the NPE method if $\delta_{1,t}, \delta_{2,t}$, and $\delta_{\mathbf{H},t}$ are small enough.

**Algorithm 4** HAQZO$^+$($\mathbf{z}_0, T, L_0, L_1, L_2, M, m, \delta$)

1: **for** $t = 0, \cdots, T-1$ **do**
2:     **if** $t \mod m = 0$ **do**
3:         Choose $\epsilon_{\mathbf{H}} > 0$
4:         $\tilde{\mathbf{H}} = \texttt{QuantumHessian}(f, \epsilon_{\mathbf{H},t}, L_0, L_1, L_2, \mathbf{z}_t, \delta/(3T))$   and   $\mathbf{H} = \mathbf{J}\tilde{\mathbf{H}}$
5:     **end if**
6:     Choose $\epsilon_{1,t} > 0$
7:     $\tilde{\mathbf{g}}_t = \texttt{QuantumGradient}(f, \epsilon_{1,t}, L_0, L_1, \mathbf{z}_t, \delta/(3T))$   and   $\mathbf{g}_t = \mathbf{J}\tilde{\mathbf{g}}_t$
8:     Compute the inexact cubic step *i.e.* find $\mathbf{z}_{t+1/2}$ that satisfies

$$\mathbf{g}_t + \left(\mathbf{H} + 6(M\|\mathbf{z}_t - \mathbf{z}_{t+1/2}\| + \sqrt{1500}d^{1/2}L_1^{1/4}\epsilon_{1,t}^{1/4} + \sqrt{1500}d^2 L_1^{1/4}L_2^{1/2}\epsilon_{\mathbf{H}}^{1/4})\mathbf{I}\right)(\mathbf{z}_{t+1/2} - \mathbf{z}_t) = \mathbf{0}$$

9:     $\lambda_t = 6\left(M\|\mathbf{z}_t - \mathbf{z}_{t+1/2}\| + \sqrt{1500}d^{1/2}L_1^{1/4}\epsilon_{1,t}^{1/4} + \sqrt{1500}d^2 L_1^{1/4}L_2^{1/2}\epsilon_{\mathbf{H}}^{1/4}\right).$
10:    Choose $\epsilon_{2,t} > 0$
11:    $\tilde{\mathbf{v}}_{t+1/2} = \texttt{QuantumGradient}(f, \epsilon_{2,t}, L_0, L_1, \mathbf{z}_{t+1/2}, \delta/(3T))$   and   $\mathbf{v}_t = \mathbf{J}\tilde{\mathbf{v}}_t$
12:    $\mathbf{z}_{t+1} = \mathbf{z}_t - \lambda_t^{-1}\mathbf{v}_t.$
13: **end for**
14: **return** $\bar{\mathbf{z}}_T = \frac{1}{\sum_{t=0}^{T-1}\lambda_t^{-1}}\sum_{t=0}^{T-1}\lambda_t^{-1}\mathbf{z}_{t+1/2}.$

---

**Lemma 4.1.** *Under Assumptions 2.1 and 2.2, let $R = \Omega(\|\mathbf{z}_0 - \mathbf{z}^*\|)$, $\{\mathbf{z}_{t+1/2}\}_{t=0}^{T-1}$ generated from (6) where $\lambda_t = 6\left(L_2\|\mathbf{z}_{t+1/2} - \mathbf{z}_t\| + \delta_{\mathbf{H},t} + \sqrt{\delta_{1,t}}\right)$, and $\delta_{1,t}$, $\delta_{2,t}$, and $\delta_{\mathbf{H},t}$ in (7) satisfy $\delta_{1,t} \leq \frac{R^2}{10T}$, $\delta_{2,t} \leq \min\left\{\frac{\lambda_t R^2}{10T(\|\mathbf{z}_{t+1/2} - \mathbf{z}_0\| + R)}, \frac{\delta_{1,t}}{2}\right\}$, $\delta_{\mathbf{H},t} \leq \frac{R}{\sqrt{T}}$, then we have $\mathrm{Gap}(\bar{\mathbf{z}}_T; \sqrt{3}R) = \mathcal{O}\left(\frac{L_2 R^3}{T^{3/2}}\right)$ where $\bar{\mathbf{z}}_T = \frac{1}{\sum_{t=0}^{T-1}\lambda_t^{-1}}\sum_{t=0}^{T-1}\lambda_t^{-1}\mathbf{z}_{t+1/2}.$*

*Remark* 4.2. We note that some prior works have also studied the inexact NPE methods [31, 1]. However, these methods only consider the case where the Hessian is inexact, while our Lemma 4.1 allows inexactness from both the gradient and the Hessian.

Because the iteration rule of Algorithm 3 can be interpreted as the generalized NPE update in (6) with high probability, we can determine the query complexity of Algorithm 3 by incorporating the quantum gradient and Hessian estimators. This result is formally stated in the following theorem.

**Theorem 4.3.** *Under Assumptions 2.1 and 2.2, let $R = \Omega(\|\mathbf{z}_0 - \mathbf{z}^*\|)$, given desired accuracy $\epsilon > 0$, we run Algorithm 3 with*

$$T = \left\lceil(3456^{1/3}L_2^{2/3} + 4^{2/3})R^2\epsilon^{-2/3}\right\rceil, \quad \epsilon_{1,t} = \frac{R^4}{15000^2 d^2 L_1 T^2}, \quad \epsilon_{\mathbf{H},t} = \frac{R^4}{1500^2 d^8 L_1 L_2^2 T^2}$$

$$\epsilon_{2,t} = \min\left\{\frac{\lambda_t R^4}{15000^2 T^2 d^2 L_1(\|\mathbf{z}_{t+1/2} - \mathbf{z}_0\| + R)^2}, \frac{\epsilon_{1,t}^2}{4}\right\}, \quad and \quad \delta \in (0,1),$$

*then with probability at $1 - \delta$, Algorithm 3 finds the $\epsilon$-saddle point of $f(\cdot)$ with $\tilde{\mathcal{O}}(dL_2^{2/3}R^2\epsilon^{-2/3})$ queries to $\mathbf{U}_f$, where $\tilde{\mathcal{O}}(\cdot)$ hides the polylogarithm dependency on $d$, $L_0$, $L_1$, $L_2$, $\epsilon^{-1}$, $\delta^{-1}$, and $R$.*

## 4.2 Hessian-Aware Quantum Algorithm with Better Dependency on $\epsilon^{-1}$ and $d$

In this section, we further improve the query complexity of HAQZO by proposing a double-loop Hessian-aware quantum method HAQZO$^+$ in Algorithm 4, which is inspired by the recent advance in lazy Hessian methods [14, 12, 10, 34].

The main difference between Algorithm 4 and Algorithm 3 is that we eliminate calling quantumHessian in every iteration, but only call it at the snapshot point in iterations $t$ when $t \mod m = 0$, and reuse such Hessian estimator in the next $m$ iterations. In addition, we replace $L_2$ with a larger parameter $M \geq L_2$ on Line 9 and Line 10 and tune it to guarantee convergence. We

**Algorithm 5** Restart-HAQZO$^+$($\mathbf{z}_0, T, L_0, L_1, L_2, M, m, S, \delta$)

1: $\mathbf{z}^{(0)} = \mathbf{z}_0$
2: **for** $s = 0, \cdots, S - 1$
3:    $\mathbf{z}^{(s+1)} = $ HAQZO$^+$($\mathbf{z}^{(s)}, T, L_0, L_1, L_2, M, m, \delta/S$)
4: **end for**
5: **return** $\mathbf{z}^{(S)}$

first consider the following iteration rule

$$\begin{cases} \mathbf{z}_{t+1/2} = \mathbf{z}_t - (\mathbf{H}_{\pi(t)} + \lambda_t \mathbf{I})^{-1} \mathbf{g}_t \\ \mathbf{z}_{t+1} = \mathbf{z}_t - \lambda_t^{-1} \mathbf{v}_t \end{cases}, \tag{8}$$

where $\pi(t) \triangleq t - (t \mod m)$ and $\mathbf{g}_t, \mathbf{v}_t, \mathbf{H}_{\pi(t)}$ are some approximations to $\mathbf{F}(\mathbf{z}_t)$, $\mathbf{F}(\mathbf{z}_{t+1/2})$, $\nabla \mathbf{F}(\mathbf{z}_{\pi(t)})$ such that

$$\|\mathbf{g}_t - \mathbf{F}(\mathbf{z}_t)\| \leq \delta_{1,t}, \quad \|\mathbf{v}_t - \mathbf{F}(\mathbf{z}_{t+1/2})\| \leq \delta_{2,t}, \quad \text{and} \quad \|\mathbf{H}_{\pi(t)} - \nabla \mathbf{F}(\mathbf{z}_{\pi(t)})\| \leq \delta_{\mathbf{H}}. \tag{9}$$

The following lemma shows that the update of (8) still enjoys the rate of $T^{-3/2}$ if the regularization term $\lambda_t$ is chosen large enough and $\delta_{1,t}, \delta_{2,t}$, and $\delta_{\mathbf{H}}$ are small.

**Lemma 4.4.** *Under Assumpions 2.1 and 2.2, let* $R = \Omega(\|\mathbf{z}_0 - \mathbf{z}^*\|)$, $\{\mathbf{z}_{t+1/2}\}_{t=0}^{T-1}$ *generated from* (8) *where* $\lambda_t = 6 \left( M \|\mathbf{z}_{t+1/2} - \mathbf{z}_t\| + \delta_{\mathbf{H},t} + \sqrt{\delta_{1,t}} \right)$, *and* $\delta_{1,t}, \delta_{2,t}, \delta_{\mathbf{H}}$ *in (9) satisfies* $\delta_{1,t} \leq \frac{R^2}{10T}$, $\delta_{2,t} \leq \min \left\{ \frac{\lambda_t R^2}{10T(\|\mathbf{z}_{t+1/2} - \mathbf{z}_0\| + R)}, \frac{\delta_{1,t}}{2} \right\}$, $\delta_{\mathbf{H}} \leq \frac{R}{\sqrt{T}}$, *if* $M \geq \frac{mL_2}{\sqrt{3}}$, *then we have* $\mathrm{Gap}(\bar{\mathbf{z}}_T; \sqrt{3}R) = \mathcal{O}\left( \frac{MR^3}{T^{3/2}} \right)$ *where* $\bar{\mathbf{z}}_T = \frac{1}{\sum_{t=0}^{T-1} \lambda_t^{-1}} \sum_{t=0}^{T-1} \lambda_t^{-1} \mathbf{z}_{t+1/2}$.

The iteration rule of Algorithm 4 can also be interpreted as (8) with high probability. At each iteration, the algorithm calls `QuantumGradient` with $\widetilde{\mathcal{O}}(1)$ quantum function evaluation queries to obtain $\mathbf{g}_t$ and $\mathbf{v}_t$. Every $m$ iterations, the algorithm calls `QuantumHessian` with $\widetilde{\mathcal{O}}(d)$ quantum function evaluation queries to obtain $\mathbf{H}_t$. The following theorem provides the query complexity of Algorithm 4 with a proper choice of $m = d$.

**Theorem 4.5.** *Under Assumptions 2.1 and 2.2, let* $R = \Omega(\|\mathbf{z}_0 - \mathbf{z}^*\|)$, *given desired accuracy* $\epsilon > 0$, *we run Algorithm 4 with*

$$m = d, \quad M = dL_2/\sqrt{3}, \quad T = \left\lceil (3456^{1/3} M^{2/3} + 4^{2/3}) R^2 \epsilon^{-2/3} \right\rceil, \quad \epsilon_{1,t} = \frac{R^4}{15000^2 d^2 L_1 T^2},$$

$$\epsilon_{\mathbf{H}} = \frac{R^4}{1500^2 d^8 L_1 L_2^2 T^2}, \quad \epsilon_{2,t} = \min \left\{ \frac{\lambda_t R^4}{15000^2 T^2 d^2 L_1 (\|\mathbf{z}_{t+1/2} - \mathbf{z}_0\| + R)^2}, \frac{\epsilon_{1,t}^2}{4} \right\}, \quad \delta \in (0, 1),$$

*then Algorithm 4 finds the* $\epsilon$-*saddle point of* $f(\cdot)$ *with* $\tilde{\mathcal{O}}(d + d^{2/3} L_2^{2/3} R^2 \epsilon^{-2/3})$ *queries to* $\mathbf{U}_f$ *with probability at* $1 - \delta$, *where* $\tilde{\mathcal{O}}(\cdot)$ *hides the polylogarithm dependency on* $d$, $L_0$, $L_1$, $L_2$, $\epsilon^{-1}$, $\delta^{-1}$, *and* $R$.

### 4.3 Restarted Hessian-Aware Quantum Algorithm for Strongly-Convex Strongly-Concave Minimax Optimization

In this section, we generalize our results to solve strongly-convex-strongly-concave minimax problems. We make the following assumption on $f(\cdot)$, which is stronger than Assumption 2.1.

**Assumption 4.6.** We assume $f(\mathbf{z}) = h(\mathbf{x}, \mathbf{y})$ is $\mu$-strongly-convex in $\mathbf{x}$ and $\mu$-strongly-concave in $\mathbf{y}$ for some $\mu > 0$.

We apply the restart strategy which is widely used in minimization [17] and minimax optimization [22, 30, 12] on our HAQZO$^+$ and propose Restart-HAQZO$^+$ in Algorithm 5. To avoid confusion, we use the superscript $^{(s)}$ to denote the parameters in the HAQZO$^+$ subroutine in the $s$-th iteration of Algorithm 5. The following lemma shows that by properly choosing the parameter in HAQZO$^+$, $\|\mathbf{z}^{(s+1)} - \mathbf{z}^*\|^2$ will descend linearly with high probability.

**Lemma 4.7.** *Under Assumptions 2.2 and 4.6, set the parameter in subroutine* HAQZO$^+$ *in the $s$-th iteration of Algorithm 5 as follows:*

$$M = \frac{mL_2}{\sqrt{3}}, \quad T = \left\lceil \frac{(100M + 12)^{2/3}\|\mathbf{z}^{(0)} - \mathbf{z}^*\|^{2/3}}{\mu^{2/3}} \right\rceil, \quad \epsilon_{1,t}^{(s)} = \frac{\|\mathbf{z}^{(s)} - \mathbf{z}^*\|^4}{1500^2 d^2 L_1^2 T^2},$$

$$\epsilon_{\mathbf{H}}^{(s)} = \frac{\|\mathbf{z}^{(s)} - \mathbf{z}^*\|^4}{1500^2 d^8 L_1 L_2^2 T^2}, \quad \epsilon_{2,t}^{(s)} = \min\left\{ \frac{(\lambda_t^{(s)})^2 \|\mathbf{z}^{(s)} - \mathbf{z}^*\|^4}{24000^2 T^2 d^2 L_1 \|\mathbf{z}_{t+1/2}^{(s)} - \mathbf{z}^*\|}, \frac{\epsilon_{1,t}^{(s)}}{4} \right\}, \quad \delta \in (0, 1),$$

*then $\|\mathbf{z}^{(s+1)} - \mathbf{z}^*\| \leq \frac{1}{2}\|\mathbf{z}^{(s)} - \mathbf{z}^*\|$ holds with probability at least $(1 - \delta/S)$.*

Lemma 4.7 means it is enough to set $S = \lceil \log(1/\epsilon) \rceil$ to obtain some $\mathbf{z}^{(S)}$ such that $\|\mathbf{z}^{(S)} - \mathbf{z}^*\|^2 \leq \epsilon$. Given this, we are ready to present the query complexity of Algorithm 5.

**Theorem 4.8.** *Under Assumptions 2.2 and 4.6, set the parameter in subroutine* HAQZO$^+$ *in the $s$ iteration of Algorithm 5 as in Lemma 4.7 with $m = d$, and set $S = \lceil \log(\|\mathbf{z}^{(0)} - \mathbf{z}^*\|^2/\epsilon) \rceil$, then with probability at least $1 - \delta$, the output of Algorithm 5 satisfies that $\|\mathbf{z}^{(S)} - \mathbf{z}^*\|^2 \leq \epsilon$ with $\tilde{\mathcal{O}}(d + d^{2/3}L_2^{2/3}\mu^{-2/3})$ queries to $\mathbf{U}_f$, where $\tilde{\mathcal{O}}(\cdot)$ hides the polylogarithm dependency on $d$, $L_0$, $L_1$, $L_2$, $\epsilon^{-1}$, $\delta^{-1}$.*

# 5 Extension to Non-convex Optimization

In the previous section, we have shown that, using quantum Hessian estimators, it is possible to design fast quantum algorithms which outperform the classical algorithms in terms of accuracy $\epsilon^{-1}$ and dimension $d$. We highlight that the quantum estimators designed in Section 3 are not restricted to convex-concave minimax problems. In this section, we extend the idea of designing Hessian-aware quantum zeroth-order methods to non-convex problems

$$\min_{\mathbf{z} \in \mathbb{R}^d} f(\mathbf{z}), \tag{10}$$

where $f(\cdot)$ is smooth but possibly not convex. We aim to find the $\epsilon$-stationary point of (10).

**Definition 5.1.** *We say $\mathbf{z}$ is an $\epsilon$-stationary point of the nonconvex minimization problem (10) if it holds that $\|\nabla f(\mathbf{z})\| \leq \epsilon$.*

We present the quantum cubic-regularized Newton methods in Algorithm 6, which replace the classical gradient and Hessian estimators in the zeroth-order CNM method [13] by the quantum estimators designed in Section 3. The following theorem gives the query complexity of Algorithm 6 to find the $\epsilon$-stationary point of $f(\cdot)$.

**Theorem 5.2.** *Under Assumption 2.2 and suppose $f^* \triangleq \min_{\mathbf{z} \in \mathbb{R}^d} f(\mathbf{z}) > -\infty$, given desired accuracy $\epsilon > 0$, we run Algorithm 6 with*

$$m = d, \quad M = 30L_2 d, \quad T = \left\lceil \frac{192M^{1/2}(f(\mathbf{x}_0) - f^*)\epsilon^{-3/2}}{3} \right\rceil,$$

$$\epsilon_{\mathbf{g}} = \frac{\epsilon^{-2}}{2000^4 d^2 L_1^2}, \quad \epsilon_{\mathbf{H}} = \frac{\epsilon^{-2}}{1500^3 d^6 L_1}, \quad and \quad \delta \in (0, 1),$$

*then with probability at least $1 - \delta$, the output of Algorithm 6 finds the $\epsilon$-stationary point of problem 10 with $\tilde{\mathcal{O}}(d + d^{1/2}L_2^{1/2}(f(\mathbf{z}_0) - f^*)\epsilon^{-3/2})$ queries to $\mathbf{U}_f$, where $\tilde{\mathcal{O}}(\cdot)$ hides the polylogarithm dependency on $d$, $L_0$, $L_1$, $L_2$, $\epsilon^{-1}$, $\delta^{-1}$.*

# 6 Conclusion

In this paper, we have proposed quantum algorithms to speed up training for minimax optimization problems. Our Hessian-aware quantum zeroth-order method reduces the query complexity of the function evaluation oracle of the classical methods by a factor of $d^{1/3}\epsilon^{-1/3}$ and $d^{1/3}\mu^{-1/3}$ for convex-concave and strongly-convex-strongly-concave problems, respectively. Moreover, we find that the proposed quantum oracles for estimating the Hessian matrix can be used to solve other important optimization problems, i.e. non-convex optimization. However, the query complexity of the proposed Hessian-aware quantum zeroth-order methods still depends on the dimension, and the quantum lower bound for this question is still unknown. We leave this for future work.

**Algorithm 6** QCNM($\mathbf{z}_0, T, L_0, L_1, L_2, M, m, \epsilon_{\mathbf{g}}, \epsilon_{\mathbf{H}}, \delta$)

1: $\delta_{\mathbf{g}} = 1500dL_1^{1/2}\epsilon_{\mathbf{g}}^{1/2}$, $\delta_{\mathbf{H}} = 1500^{1/2}d^2L_1^{1/4}L_2^{1/2}\epsilon_{\mathbf{H}}^{1/4}$
2: **for** $t = 0, \cdots, T-1$ **do**
3:    **if** $t \mod m = 0$ **do**
4:       $\mathbf{H} = \texttt{QuantumHessian}(f, \epsilon_{\mathbf{H}}, L_0, L_1, L_2, \mathbf{z}_t, \delta/(2T))$
5:    **end if**
6:    $\mathbf{g}_t = \texttt{QuantumGradient}(f, \epsilon_{\mathbf{g}}, L_0, L_1, \mathbf{z}_t, \delta/(2T))$
7:    Compute the cubic step *i.e.* find $\mathbf{z}_{t+1}$ that satisfies

$$\mathbf{z}_{t+1} = \arg\min_{\mathbf{z} \in \mathbb{R}^d} \left\{ \langle \mathbf{g}_t, \mathbf{z} - \mathbf{z}_t \rangle + \frac{1}{2}\langle \mathbf{H} \cdot (\mathbf{z} - \mathbf{z}_t), \mathbf{z} - \mathbf{z}_t \rangle + \frac{M}{6}\|\mathbf{z} - \mathbf{z}_t\|^3 \right\}$$

8: **end for**
9: **return** $\mathbf{z}_{\text{out}}$ uniformly from $\{\mathbf{z}_i\}_{i=1}^T$

## Acknowledgment

We thank the anonymous reviewers for their helpful suggestions. Zongqi Wan, Jialin Zhang, and Xiaoming Sun are supported by the National Natural Science Foundation of China Grants No. 62325210 and 12447107. Chengchang Liu is supported by the National Natural Science Foundation of China (624B2125). John C.S. Lui is supported in part by the GRF-14207721 and SRFS2122-4S02.

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

## A   Auxiliary Lemmas

**Lemma A.1.** *Given a positive sequence $\{\lambda_t\}_{t=0}^{T-1}$, if $\sum_{t=0}^{T-1} \lambda_t^2 \leq C$, then we have $\sum_{t=0}^{T-1} \frac{1}{\lambda_t} \geq \frac{T^{3/2}}{\sqrt{C}}$.*

*Proof.* By Holder's inequality, we have that

$$T = \sum_{t=0}^{T-1} \left(\frac{1}{\lambda_t}\right)^{2/3} (\lambda_t^2)^{1/3} \leq \left(\sum_{t=0}^{T-1} \frac{1}{\lambda_t}\right)^{2/3} \left(\sum_{t=0}^{T-1} \lambda_t^2\right)^{1/3}.$$

Therefore,

$$\sum_{t=0}^{T-1} \frac{1}{\lambda_t} \geq \frac{T^{3/2}}{\sqrt{C}}. \tag{11}$$

$\square$

**Lemma A.2** (Lemma 4.2, [12]). *For any sequence of positive numbers $\{r_t\}_{t\geq 0}$, it holds for any $m \geq 2$ that $\sum_{t=1}^{m-1} \left(\sum_{i=0}^{t-1} r_i\right)^2 \leq \frac{m^2}{2} \sum_{t=0}^{m-1} r_t^2$.*

**Lemma A.3** (Lemma 4.2, [14]). *For any sequence of positive numbers $\{r_t\}_{t\geq 0}$, it holds for any $m \geq 1$ that $\sum_{t=1}^{m-1} \left(\sum_{i=0}^{t-1} r_i\right)^3 \leq \frac{m^3}{3} \sum_{t=0}^{m-1} r_t^3$.*

# B The Proof of Section 3

## B.1 The Proof of Lemma 3.2

*Proof.* Running $\mathcal{A}(f, \epsilon_{\mathbf{g}}, L_0, L_1, \mathbf{x})$ for $M$ times. Let $\tilde{\nabla}^{(m)} f(\mathbf{z})$ denotes the output estimates for $m$-th running. Then for each coordination, take the median of the output estimates, and return the resulting vector as the output of $\texttt{QuantumGradient}(f, \epsilon_{\mathbf{g}}, L_0, L_1, \mathbf{z}, \delta)$, denoted as $\tilde{\nabla} f(\mathbf{z})$.

For any $i \in [d], m \in [M]$, $X_{i,m}$ denotes the indicator random variable of the event
$$\left| \left[ \tilde{\nabla}^{(m)} f(\mathbf{z}) \right]_i - [\nabla f(\mathbf{z})]_i \right| < 1500\sqrt{L_1 d \epsilon_{\mathbf{g}}}.$$
By Lemma 3.1, we have $\Pr(X_{m,i}) \geq \frac{2}{3}$ and $\{X_{m,i}\}_{m=1}^M$ are independent random variables. By Chernoff's bound,
$$\Pr\left( \sum_{m=1}^M X_{i,m} \leq \frac{5}{9}M \right) \leq e^{-\frac{13M}{500}}.$$
Let $Y_i$ denote the event that $\sum_{m=1}^M X_{i,m} \geq \frac{5}{9}M$, then $\Pr(Y_i) \geq 1 - e^{-\frac{13M}{500}}$. If $Y_i$ happens, we have $\left| [\tilde{\nabla} f(vz)]_i - [\nabla f(\mathbf{z})]_i \right| \leq 1500\sqrt{L_1 d \epsilon_{\mathbf{g}}}$. By union bound, $\Pr(\cap_{i=1}^d Y_i) \geq 1 - d \cdot e^{-\frac{13M}{500}}$. Under event $\cap_{i=1}^d Y_i$, we have $\|\tilde{\nabla} f(\mathbf{z}) - \nabla f(\mathbf{z})\|_2 \leq \sqrt{1500^2 d \cdot L_1 d \epsilon_{\mathbf{g}}} = 1500d\sqrt{L_1 \epsilon_{\mathbf{g}}}$. Set $M := \mathcal{O}(\log(\frac{d}{\delta}))$, we have $\Pr(\cap_{i=1}^d Y_i) \geq 1 - \delta$.

Since we have invoked $\mathcal{A}$ for $M$ times, the total query complexity is $\mathcal{O}(\log(\frac{d}{\delta}))$ and the total gate complexity is $\mathcal{O}(d \log \frac{L_0}{dL_1 \epsilon_{\mathbf{g}}} \log \frac{d}{\delta})$ by Lemma 3.1. $\square$

## B.2 The Proof of Lemma 3.3

*Proof.* Let $\tilde{\nabla} f(\mathbf{x})$ and $\tilde{\nabla} f(\mathbf{x} + \Delta)$ be the quantum gradient estimates with probability at least $(1 - \delta/2)$. By Lemma 3.2 and union bound, we have
$$\|\tilde{\nabla} f(\mathbf{x}) - \nabla f(\mathbf{x})\|_2 \leq 1500d\sqrt{L_1 \epsilon_{\mathbf{hv}}} \quad \text{and} \quad \|\tilde{\nabla} f(\mathbf{x} + \Delta) - \nabla f(\mathbf{x} + \Delta)\|_2 \leq 1500d\sqrt{L_1 \epsilon_{\mathbf{hv}}}$$
hold with probability at least $1 - \delta$. Condition on this good event, we have
$$
\begin{aligned}
\left\| \mathbf{hv} - \nabla^2 f(\mathbf{x})\mathbf{v} \right\|_2 &\leq \left\| \frac{1}{\Delta} \left( \nabla f(\mathbf{x} + \Delta\mathbf{v}) - \nabla f(\mathbf{x}) \right) - \nabla^2 f(\mathbf{x})\mathbf{v} \right\|_2 \\
&\quad + \frac{1}{\Delta} \|\tilde{\nabla} f(\mathbf{x} + \Delta\mathbf{v}) - \nabla f(\mathbf{x} + \Delta\mathbf{v})\|_2 + \frac{1}{\Delta} \|\tilde{\nabla} f(\mathbf{x}) - \nabla f(\mathbf{x})\|_2 \\
&\leq \frac{L_2 \Delta}{2} \|\mathbf{v}\|_2^2 + \frac{3000d\sqrt{L_1 \epsilon_{\mathbf{hv}}}}{\Delta}.
\end{aligned}
$$
Since we have choose $\Delta = 20\sqrt{15}\epsilon_{\mathbf{hv}}^{1/4} M^{-1/2} L_2^{-1/2} L_1^{1/4} d^{1/2}$, it holds that
$$\Pr\left( \left\| \mathbf{hv} - \nabla^2 f(\mathbf{x})\mathbf{v} \right\|_2 \leq 10\sqrt{15}(dL_2 M)^{1/2} (\epsilon_{\mathbf{hv}} L_1)^{1/4} \right) \geq 1 - \delta. \tag{12}$$
$\square$

## B.3 The Proof of Lemma 3.5

*Proof.* It holds that $\mathbf{H}[i, :]$ estimates $\nabla^2 f(\mathbf{z})\mathbf{e}_i$ with failure probability $\delta/d$, then by Lemma 3.3,
$$\left\| \mathbf{H}[i, :] - \nabla^2 f(\mathbf{z})\mathbf{e}_i \right\|_2 \leq 10\sqrt{15}(dL_2)^{1/2} (\epsilon L_1)^{1/4}, \quad \text{for all} \quad i \in [d]$$
holds with probability $1 - \delta$. Under this event, we have
$$\left\| \mathbf{H} - \nabla^2 f(\mathbf{z}) \right\|_F^2 \leq \sum_{i=1}^d \|\mathbf{H}[i, :] - \nabla^2 f(\mathbf{z})\mathbf{e}_i\|_2^2 \leq 10\sqrt{15}d^{3/2} L_2^{1/2} (\epsilon_{\mathbf{H}} L_1)^{1/4}.$$
Therefore, we have
$$\left\| \tilde{\mathbf{H}} - \nabla^2 f(\mathbf{z}) \right\|_2 \leq \left\| \mathbf{H} - \nabla^2 f(\mathbf{z}) \right\|_2 \leq \sqrt{d} \left\| \mathbf{H} - \nabla^2 f(\mathbf{z}) \right\|_F \leq 10\sqrt{15}d^2 L_2^{1/2} (\epsilon_{\mathbf{H}} L_1)^{1/4}$$
holds with probability at least $1 - \delta$. $\square$

## C  The Proof of Section 4

### C.1  The Proof of Lemma 4.1

*Proof.* The iteration rule (6) means

$$\mathbf{g}_t + \mathbf{H}_t(\mathbf{z}_{t+1/2} - \mathbf{z}_t) + \lambda_t(\mathbf{z}_{t+1/2} - \mathbf{z}_t) = \mathbf{0}. \tag{13}$$

Thus, we have

$$\left\| \mathbf{z}_{t+1/2} - (\mathbf{z}_t - \frac{1}{\lambda_t}\mathbf{v}_t) \right\| = \left\| \mathbf{z}_{t+1/2} - \left( \mathbf{z}_t - \frac{1}{\lambda_t}\mathbf{F}(\mathbf{z}_{t+1/2}) \right) \right\| + \frac{1}{\lambda_t} \left\| \mathbf{v}_t - \mathbf{F}(\mathbf{z}_{t+1/2}) \right\|$$

$$\overset{(7)}{\leq} \frac{\delta_{2,t}}{\lambda_t} + \left\| \mathbf{z}_{t+1/2} - \mathbf{z}_t + \frac{1}{\lambda_t}(\mathbf{g}_t + \mathbf{H}_t(\mathbf{z}_{t+1/2} - \mathbf{z}_t)) \right\| + \left\| \frac{1}{\lambda_t}(\mathbf{F}(\mathbf{z}_{t+1/2}) - \mathbf{g}_t - \mathbf{H}_t(\mathbf{z}_{t+1/2} - \mathbf{z}_t)) \right\|$$

$$\overset{(13),(2)}{\leq} \frac{\delta_{2,t}}{\lambda_t} + \frac{L_2\|\mathbf{z}_{t+1/2} - \mathbf{z}_t\|^2}{2\lambda_t} + \frac{\|\mathbf{g}_t - \mathbf{F}(\mathbf{z}_t)\|}{\lambda_t} + \frac{\|\mathbf{H}_t - \nabla\mathbf{F}(\mathbf{z}_t)\|\|\mathbf{z}_{t+1/2} - \mathbf{z}_t\|}{\lambda_t}$$

$$\overset{(7)}{\leq} \frac{1}{\lambda_t}\left( L_2\|\mathbf{z}_{t+1/2} - \mathbf{z}_t\|^2 + \frac{3}{2}\delta_{1,t} + \delta_{\mathbf{H},t}\|\mathbf{z}_{t+1/2} - \mathbf{z}_t\| \right)$$

$$\leq \frac{1}{6}\|\mathbf{z}_{t+1/2} - \mathbf{z}_t\| + \frac{1}{4}\sqrt{\delta_{1,t}}, \tag{14}$$

where the last inequality is due to the choice of $\lambda_t$. We also have that

$$\frac{1}{\lambda_t}\langle \mathbf{v}_t, \mathbf{z}_{t+1/2} - \mathbf{z} \rangle = \langle \mathbf{z}_t - \mathbf{z}_{t+1}, \mathbf{z}_{t+1/2} - \mathbf{z} \rangle$$

$$= \langle \mathbf{z}_t - \mathbf{z}_{t+1}, \mathbf{z}_{t+1} - \mathbf{z} \rangle + \langle \mathbf{z}_t - \mathbf{z}_{t+1/2}, \mathbf{z}_{t+1/2} - \mathbf{z}_{t+1} \rangle + \langle \mathbf{z}_{t+1/2} - \mathbf{z}_{t+1}, \mathbf{z}_{t+1/2} - \mathbf{z}_{t+1} \rangle \tag{15}$$

$$\leq \frac{1}{2}\left( \|\mathbf{z}_t - \mathbf{z}\|^2 - \|\mathbf{z}_{t+1} - \mathbf{z}\|^2 \right) + \frac{1}{2}\|\mathbf{z}_{t+1/2} - \mathbf{z}_{t+1}\|^2 - \frac{1}{2}\|\mathbf{z}_t - \mathbf{z}_{t+1/2}\|^2.$$

Since $\|\mathbf{z}_{t+1/2} - (\mathbf{z}_t - \frac{1}{\lambda_t}\mathbf{v}_t)\| = \|\mathbf{z}_{t+1/2} - \mathbf{z}_{t+1}\|$, plugging the bound of (14) into (15) and using the fact that $\|\mathbf{v}_t - \mathbf{F}(\mathbf{z}_{t+1/2})\| \leq \delta_{2,t}$, we have

$$\frac{1}{\lambda_t}\langle \mathbf{F}(\mathbf{z}_{t+1/2}), \mathbf{z}_{t+1/2} - \mathbf{z} \rangle + \frac{1}{4}\|\mathbf{z}_{t+1/2} - \mathbf{z}_t\|^2$$

$$= \frac{1}{\lambda_t}\langle \mathbf{v}_t, \mathbf{z}_{t+1/2} - \mathbf{z} \rangle + \frac{1}{4}\|\mathbf{z}_{t+1/2} - \mathbf{z}_t\|^2 + \frac{1}{\lambda_t}\langle \mathbf{F}(\mathbf{z}_{t+1/2}) - \mathbf{v}_t, \mathbf{z}_{t+1/2} - \mathbf{z} \rangle$$

$$\overset{(15)}{\leq} \frac{1}{2}\left( \|\mathbf{z}_t - \mathbf{z}\|^2 - \|\mathbf{z}_{t+1} - \mathbf{z}\|^2 \right) + \frac{1}{4}\|\mathbf{z}_{t+1/2} - \mathbf{z}_t\|^2$$

$$+ \frac{1}{2}\|\mathbf{z}_{t+1/2} - \mathbf{z}_{t+1}\|^2 - \frac{1}{2}\|\mathbf{z}_t - \mathbf{z}_{t+1}\|^2 + \delta_{2,t}\frac{\|\mathbf{z}_{t+1/2} - \mathbf{z}\|}{\lambda_t} \tag{16}$$

$$\overset{(14)}{\leq} \frac{1}{2}\left( \|\mathbf{z}_t - \mathbf{z}\|^2 - \|\mathbf{z}_{t+1} - \mathbf{z}\|^2 \right) + \frac{1}{4}\|\mathbf{z}_{t+1/2} - \mathbf{z}_t\|^2$$

$$+ \frac{1}{2}\left( \frac{1}{18}\|\mathbf{z}_{t+1/2} - \mathbf{z}_t\|^2 + \frac{1}{8}\delta_{1,t} \right) - \frac{1}{2}\|\mathbf{z}_t - \mathbf{z}_{t+1/2}\|^2 + \delta_{2,t}\frac{\|\mathbf{z}_{t+1/2} - \mathbf{z}\|}{\lambda_t}$$

$$\leq \frac{1}{2}\left( \|\mathbf{z}_t - \mathbf{z}\|^2 - \|\mathbf{z}_{t+1} - \mathbf{z}\|^2 \right) + \delta_{1,t} + \delta_{2,t}\frac{\|\mathbf{z}_{t+1/2} - \mathbf{z}\|}{\lambda_t}.$$

We let $R \geq 10\|\mathbf{z}_0 - \mathbf{z}^*\|$ and the choice of $\delta_{1,t}$ and $\delta_{2,t}$ means that we have

$$\delta_{1,t} + \delta_{2,t}\frac{\|\mathbf{z}_{t+1/2} - \mathbf{z}^*\|}{\lambda_t} \leq \delta_{1,t} + \delta_{2,t}(\|\mathbf{z}_{t+1/2} - \mathbf{z}_0\| + \|\mathbf{z}_0 - \mathbf{z}^*\|)$$

$$\leq \frac{R^2}{10T} + \frac{R^2(R + \|\mathbf{z}_{t+1/2} - \mathbf{z}_0\|)}{10T(\|\mathbf{z}_{t+1/2} - \mathbf{z}_0\| + R)} \leq \frac{R^2}{5T}.$$

Let $\mathbf{z} = \mathbf{z}^*$ in (16) and due to $\langle \mathbf{F}(\mathbf{z}_{k+1/2}), \mathbf{z}_{k+1/2} - \mathbf{z}^* \rangle \geq 0$ for all $k$, we have

$$\|\mathbf{z}_t - \mathbf{z}^*\|^2 \leq \|\mathbf{z}_0 - \mathbf{z}^*\|^2 + 2\sum_{k=0}^{t-1}\left( \delta_{1,k} + \delta_{2,k}\frac{\|\mathbf{z}_{k+1/2} - \mathbf{z}^*\|}{\lambda_k} \right) \leq \frac{R^2}{2}$$

and

$$\|\mathbf{z}_{t+1/2} - \mathbf{z}_t\|^2 \leq 2\|\mathbf{z}_0 - \mathbf{z}^*\|^2 + 2\sum_{k=0}^{t-1}\left(\delta_{1,k} + \delta_{2,k}\frac{\|\mathbf{z}_{k+1/2} - \mathbf{z}^*\|}{\lambda_k}\right) \leq \frac{R^2}{5} + \frac{2R^2}{5} = \frac{3R^2}{5}.$$

Thus, we have

$$\|\mathbf{z}_{t+1/2} - \mathbf{z}^*\|^2 \leq 2\|\mathbf{z}_{t+1/2} - \mathbf{z}_t\|^2 + 2\|\mathbf{z}_t - \mathbf{z}^*\|^2 \leq 3R^2. \tag{17}$$

Summing up the (16) from $t = 0$ to $t = T - 1$, for all $\mathbf{z} \in \mathbb{B}_{\sqrt{6}R}(\mathbf{z}^*)$, we have

$$\sum_{t=0}^{T-1}\frac{1}{\lambda_t}\langle\mathbf{F}(\mathbf{z}_{t+1/2}), \mathbf{z}_{t+1/2} - \mathbf{z}\rangle + \sum_{t=0}^{T-1}\frac{1}{4}\|\mathbf{z}_{t+1/2} - \mathbf{z}_t\|^2$$

$$\leq \frac{1}{2}\left(\|\mathbf{z}_0 - \mathbf{z}\|^2 - \|\mathbf{z}_T - \mathbf{z}\|^2\right) + \sum_{t=0}^{T-1}\left(\delta_{1,t} + \delta_{2,t}\frac{\|\mathbf{z}_{t+1/2} - \mathbf{z}\|}{\lambda_t}\right) \tag{18}$$

$$\leq \frac{R^2}{20} + \frac{R^2}{10} + \frac{TR^2(\|\mathbf{z}_{t+1/2} - \mathbf{z}^*\| + \|\mathbf{z} - \mathbf{z}^*\|)}{10T(\|\mathbf{z}_{t+1/2} - \mathbf{z}_0\| + R)} \leq 2R^2.$$

We then bound the regularization term $\sum_{t=0}^{T-1}\lambda_t^2$ by

$$\sum_{t=0}^{T-1}\lambda_t^2 \leq \sum_{t=0}^{T-1}36(3L_2^2\|\mathbf{z}_{t+1/2} - \mathbf{z}_t\|^2 + 3\delta_{\mathbf{H},t}^2 + 3\delta_{1,t})$$

$$\leq 864L_2^2R^2 + \sum_{t=0}^{T-1}\left(\frac{3R^2}{T} + \frac{3R^2}{10T}\right) \leq (864L_2^2 + 4)R^2.$$

Thus, we have

$$\text{Gap}(\bar{\mathbf{z}}_T; \sqrt{3}R) \leq \max_{\|\mathbf{z}-\mathbf{z}^*\|\leq\sqrt{6}R}\langle\mathbf{F}(\mathbf{z}), \bar{\mathbf{z}}_T - \mathbf{z}\rangle = \max_{\|\mathbf{z}-\mathbf{z}^*\|\leq\sqrt{6}R}\frac{1}{\sum_{t=0}^{T-1}1/\lambda_t}\sum_{t=0}^{T-1}\frac{1}{\lambda_t}\langle\mathbf{F}(\mathbf{z}), \mathbf{z}_{t+1/2} - \mathbf{z}\rangle$$

$$\overset{*}{\leq} \max_{\|\mathbf{z}-\mathbf{z}^*\|\leq\sqrt{6}R}\frac{1}{\sum_{t=0}^{T-1}1/\lambda_t}\sum_{t=0}^{T-1}\frac{1}{\lambda_t}\langle\mathbf{F}(\mathbf{z}_{t+1/2}), \mathbf{z}_{t+1/2} - \mathbf{z}\rangle \overset{(18)}{\leq} \frac{2R^2}{\sum_{t=0}^{T-1}1/\lambda_t}$$

$$\leq \frac{2\sqrt{864L_2^2 + 4}R^3}{T^{3/2}},$$

where the first inequality is from Proposition 2.3, $*$ is due to the monotone of $\mathbf{F}(\cdot)$, and the last inequality is due to $\sum_{t=0}^{T-1}\frac{1}{\lambda_t} \geq \frac{T^{3/2}}{\sqrt{(864L_2^2+4)R^2}}$ according to Lemma A.1. $\qquad\square$

### C.2 The Proof of Theorem 4.3

*Proof.* According to Lemmas 3.2 and 3.5, we know that $\mathbf{g}_t$, $\mathbf{H}_t$, $\mathbf{v}_t$ can be constructed within $\tilde{\mathcal{O}}(1)$, $\tilde{\mathcal{O}}(d)$, and $\tilde{\mathcal{O}}(1)$ quantum function evaluation oracle, respectively. The following statements

$$\|\mathbf{g}_t - \mathbf{F}(\mathbf{z}_t)\| \leq \|\mathbf{J}\|\|\tilde{\mathbf{g}}_t - \nabla f(\mathbf{z}_t)\| \leq 1500d\sqrt{L_1\epsilon_{1,t}} \leq \frac{R^2}{10T},$$

$$\|\mathbf{H}_t - \nabla\mathbf{F}(\mathbf{z}_t)\| \leq \|\mathbf{J}\|\|\tilde{\mathbf{H}}_{t,t} - \nabla^2 f(\mathbf{z}_t)\| \leq 10\sqrt{15}d^2 L_1^{1/4}L_2^{1/2}\epsilon_{\mathbf{H}}^{1/4} \leq \frac{R}{\sqrt{T}},$$

and

$$\|\mathbf{v}_t - \mathbf{F}(\mathbf{z}_{t+1/2})\| \leq \|\mathbf{J}\|\|\tilde{\mathbf{v}}_t - \nabla f(\mathbf{z}_{t+1/2})\|$$

$$\leq 1500d\sqrt{L_1\epsilon_{2,t}} \leq \min\left\{\frac{\lambda_t R^2}{10T\left(\|\mathbf{z}_{t+1/2} - \mathbf{z}_0\| + R\right)}, \frac{R^2}{2T}\right\}$$

hold with probability at least $\left(1 - \frac{\delta}{T}\right)$.

Let $\delta_{1,t} = \frac{R^2}{10T}$, $\delta_{\mathbf{H},t} = \frac{R}{\sqrt{T}}$, and $\delta_{2,t} = \min\left\{\frac{\lambda_t R^2}{10T\left(\|\mathbf{z}_{t+1/2} - \mathbf{z}_0\| + R\right)}, \frac{\delta_{1,t}}{2}\right\}$, we know that the condition of Lemma 4.1 holds with probability at least $(1 - \delta)$. Thus, the output $\bar{\mathbf{z}}_T$ of Algorithm 3 holds that

$$\mathrm{Gap}(\bar{\mathbf{z}}_T; \sqrt{3}R) \leq \frac{2\sqrt{864L_2^2 + 4R^3}}{T^{3/2}} \leq \epsilon,$$

with probability at least $(1 - \delta)$. The total query of quantum evaluation oracle can be bounded by

$$\#\mathrm{Query} = \left(\tilde{\mathcal{O}}(1) + \tilde{\mathcal{O}}(1) + \tilde{\mathcal{O}}(d)\right) \cdot T = \tilde{\mathcal{O}}(dL_2^{2/3}R^2\epsilon^{-2/3}).$$

$\square$

## C.3 The Proof of Lemma 4.4

*Proof.* The iteration of (8) means

$$\mathbf{g}_t + \mathbf{H}_{\pi(t)}(\mathbf{z}_{t+1/2} - \mathbf{z}_t) + \lambda_t(\mathbf{z}_{t+1/2} - \mathbf{z}_t) = \mathbf{0}. \tag{19}$$

Thus, we have

$$\left\|\mathbf{z}_{t+1/2} - \left(\mathbf{z}_t - \frac{1}{\lambda_t}\mathbf{v}_t\right)\right\|$$

$$= \left\|\mathbf{z}_{t+1/2} - \left(\mathbf{z}_t - \frac{1}{\lambda_t}\mathbf{F}(\mathbf{z}_{t+1/2})\right)\right\| + \frac{1}{\lambda_t}\left\|\mathbf{v}_t - \mathbf{F}(\mathbf{z}_{t+1/2})\right\|$$

$$\leq \frac{\delta_{2,t}}{\lambda_t} + \underbrace{\left\|\mathbf{z}_{t+1/2} - \mathbf{z}_t + \frac{1}{\lambda_t}(\mathbf{g}_t + \mathbf{H}_{\pi(t)}(\mathbf{z}_{t+1/2} - \mathbf{z}_t))\right\|}_{=0}$$

$$+ \left\|\frac{1}{\lambda_t}(\mathbf{F}(\mathbf{z}_{t+1/2}) - \mathbf{g}_t - \mathbf{H}_{\pi(t)}(\mathbf{z}_{t+1/2} - \mathbf{z}_t))\right\|$$

$$\leq \frac{\delta_{2,t}}{\lambda_t} + \frac{1}{\lambda_t}\left\|\mathbf{F}(\mathbf{z}_{t+1/2}) - \mathbf{F}(\mathbf{z}_t) - \nabla\mathbf{F}(\mathbf{z}_t)(\mathbf{z}_{t+1/2} - \mathbf{z}_t)\right\|$$

$$+ \frac{1}{\lambda_t}\left(\|\mathbf{g}_t - \mathbf{F}(\mathbf{z}_t)\| + (\|\mathbf{H}_\pi(t) - \nabla\mathbf{F}(\mathbf{z}_{\pi(t)})\| + \|\nabla\mathbf{F}(\mathbf{z}_{\pi(t)}) - \nabla\mathbf{F}(\mathbf{z}_t)\|)\|\mathbf{z}_{t+1/2} - \mathbf{z}_t\|\right)$$

$$\overset{(2)}{\leq} \frac{\delta_{2,t}}{\lambda_t} + \frac{L_2\|\mathbf{z}_{t+1/2} - \mathbf{z}_t\|^2}{2\lambda_t} + \frac{\|\mathbf{g}_t - \mathbf{F}(\mathbf{z}_t)\|}{\lambda_t}$$

$$+ \frac{(\|\mathbf{H}_{\pi(t)} - \nabla\mathbf{F}(\mathbf{z}_{\pi(t)})\| + \|\nabla\mathbf{F}(\mathbf{z}_{\pi(t)}) - \nabla\mathbf{F}(\mathbf{z}_t)\|)\|\mathbf{z}_{t+1/2} - \mathbf{z}_t\|}{\lambda_t}$$

$$\leq \frac{\delta_{2,t}}{\lambda_t} + \frac{1}{\lambda_t}(L_2\|\mathbf{z}_{t+1/2} - \mathbf{z}_t\|^2 + \delta_{1,t} + \delta_{\mathbf{H}}\|\mathbf{z}_{t+1/2} - \mathbf{z}_t\|) + \frac{L_2}{\lambda_t}(\|\mathbf{z}_{\pi(t)} - \mathbf{z}_t\|\|\mathbf{z}_{t+1/2} - \mathbf{z}_t\|)$$

$$\leq \frac{L_2}{6M}\|\mathbf{z}_{t+1/2} - \mathbf{z}_t\| + \frac{1}{4}\sqrt{\delta_{1,t}} + \frac{L_2}{6M}\|\mathbf{z}_{\pi(t)} - \mathbf{z}_t\|, \tag{20}$$

where the last inequality is due to the choice of $\lambda_t$. On the other hand, it holds that

$$\frac{1}{\lambda_t}\langle\mathbf{v}_t, \mathbf{z}_{t+1/2} - \mathbf{z}\rangle$$

$$\leq \frac{1}{2}\left(\|\mathbf{z}_t - \mathbf{z}\|^2 - \|\mathbf{z}_{t+1} - \mathbf{z}\|^2\right) + \underbrace{\frac{1}{2}\|\mathbf{z}_{t+1/2} - \mathbf{z}_{t+1}\|^2}_{=\frac{1}{2}\|\mathbf{z}_{t+1/2} - \mathbf{z}_{t+1}\|^2 - \frac{1}{4}\|\mathbf{z}_{t+1/2} - \mathbf{z}_t\|^2} - \frac{1}{2}\|\mathbf{z}_t - \mathbf{z}_{t+1/2}\|^2 \tag{21}$$

$$\overset{(20)}{\leq} \frac{1}{2}\left(\|\mathbf{z}_t - \mathbf{z}\|^2 - \|\mathbf{z}_{t+1} - \mathbf{z}\|^2\right) + \frac{3}{16}\delta_{1,t} + \left(\frac{L_2^2}{12M^2} - \frac{1}{4}\right)\|\mathbf{z}_{t+1/2} - \mathbf{z}_t\|^2$$

$$+ \frac{L_2^2}{12M^2}\|\mathbf{z}_{\pi(t)} - \mathbf{z}_t\|^2 - \frac{1}{4}\|\mathbf{z}_{t+1/2} - \mathbf{z}_{t+1}\|^2 - \frac{1}{4}\|\mathbf{z}_t - \mathbf{z}_{t+1/2}\|^2.$$

We denote $r_t \triangleq \|\mathbf{z}_{t+1} - \mathbf{z}_t\|$, since $r_t^2 \leq 2\|\mathbf{z}_{t+1/2} - \mathbf{z}_t\|^2 + 2\|\mathbf{z}_{t+1} - \mathbf{z}_t\|^2$ and that

$$\|\mathbf{z}_{\pi(t)} - \mathbf{z}_t\| = \left\| \sum_{i=\pi(t)}^{t-1} (\mathbf{z}_{i+1} - \mathbf{z}_i) \right\| \leq \sum_{i=\pi(t)}^{t-1} \|\mathbf{z}_{i+1} - \mathbf{z}_i\| \leq \sum_{i=\pi(t)}^{t-1} r_i, \qquad (22)$$

then it holds that

$$\frac{1}{\lambda_t} \langle \mathbf{v}_t, \mathbf{z}_{t+1/2} - \mathbf{z} \rangle + \left( \frac{1}{4} - \frac{L_2^2}{12M^2} \right) \|\mathbf{z}_{t+1/2} - \mathbf{z}_t\|^2$$

$$\overset{(21),(22)}{\leq} \frac{1}{2} \left( \|\mathbf{z}_t - \mathbf{z}\|^2 - \|\mathbf{z}_{t+1} - \mathbf{z}\|^2 \right) + \frac{3\delta_{1,t}}{16} - \left( \frac{1}{8} r_t^2 - \frac{L_2^2 \left( \sum_{i=\pi(t)}^{t-1} r_i \right)^2}{12M^2} \right). \qquad (23)$$

Summing up the above inequality from $i = \pi(t)$ to $\pi(t) + s - 1$ where $1 \leq s \leq m$ and take $M \geq \frac{mL_2}{\sqrt{3}}$, we have

$$\sum_{i=\pi(t)}^{t-1} \left( \frac{1}{\lambda_i} \langle \mathbf{v}_i, \mathbf{z}_{i+1/2} - \mathbf{z} \rangle + \frac{1}{8} \|\mathbf{z}_{i+1/2} - \mathbf{z}_i\|^2 \right)$$

$$\leq \frac{1}{2} \left( \|\mathbf{z}_{\pi(t)} - \mathbf{z}\|^2 - \|\mathbf{z}_{\pi(t)+s} - \mathbf{z}\|^2 \right) + \frac{3}{16} \sum_{i=\pi(t)}^{t-1} \delta_{1,i}, \qquad (24)$$

where the last inequality is due to Lemma A.2. Combining the error from $\mathbf{v}_t$, we have that

$$\sum_{i=\pi(t)}^{t-1} \left( \frac{1}{\lambda_i} \langle \mathbf{F}(\mathbf{z}_{i+1/2}), \mathbf{z}_{i+1/2} - \mathbf{z} \rangle + \frac{1}{8} \|\mathbf{z}_{i+1/2} - \mathbf{z}_i\|^2 \right)$$

$$\leq \sum_{i=\pi(t)}^{t-1} \left( \frac{1}{\lambda_i} \langle \mathbf{v}_i, \mathbf{z}_{i+1/2} - \mathbf{z} \rangle + \frac{1}{8} \|\mathbf{z}_{i+1/2} - \mathbf{z}_i\|^2 \right) + \sum_{i=\pi(t)}^{t-1} \frac{\|\mathbf{F}(\mathbf{z}_{i+1/2}) - \mathbf{v}_i\| \|\mathbf{z}_{i+1/2} - \mathbf{z}\|}{\lambda_i}$$

$$\overset{(24)}{\leq} \frac{1}{2} \left( \|\mathbf{z}_{\pi(t)} - \mathbf{z}\|^2 - \|\mathbf{z}_t - \mathbf{z}\|^2 \right) + \sum_{i=\pi(t)}^{t-1} \left( \frac{3}{16} \delta_{1,i} + \frac{\delta_{2,i}}{\lambda_i} \|\mathbf{z}_{i+1/2} - \mathbf{z}\| \right). \qquad (25)$$

Similar to the proof in Lemma 4.1, we let $R \geq 10\|\mathbf{z}_0 - \mathbf{z}^*\|$ and the choice of $\delta_{1,t}$ and $\delta_{2,t}$ means

$$\delta_{1,t} + \delta_{2,t} \frac{\|\mathbf{z}_{t+1/2} - \mathbf{z}^*\|}{\lambda_t} \leq \delta_{1,t} + \delta_{2,t}(\|\mathbf{z}_{t+1/2} - \mathbf{z}_0\| + \|\mathbf{z}_0 - \mathbf{z}^*\|)$$

$$\leq \frac{R^2}{10T} + \frac{R^2(R + \|\mathbf{z}_{t+1/2} - \mathbf{z}_0\|)}{10T(\|\mathbf{z}_{t+1/2} - \mathbf{z}_0\| + R)} \leq \frac{R^2}{5T}. \qquad (26)$$

Let $\mathbf{z} = \mathbf{z}^*$ in (25), we have

$$\|\mathbf{z}_t - \mathbf{z}^*\|^2 \leq \|\mathbf{z}_{\pi(t)} - \mathbf{z}^*\|^2 + 2 \sum_{k=\pi(t)}^{t-1} \left( \delta_{1,k} + \delta_{2,k} \frac{\|\mathbf{z}_{k+1/2} - \mathbf{z}^*\|}{\lambda_k} \right)$$

$$\leq \|\mathbf{z}_0 - \mathbf{z}^*\|^2 + 2 \sum_{k=0}^{t-1} \left( \delta_{1,k} + \delta_{2,k} \frac{\|\mathbf{z}_{k+1/2} - \mathbf{z}^*\|}{\lambda_k} \right) \overset{(26)}{\leq} \frac{R^2}{2}. \qquad (27)$$

and

$$\|\mathbf{z}_{t+1/2} - \mathbf{z}_t\|^2 \leq 4\|\mathbf{z}_{\pi(t)} - \mathbf{z}^*\|^2 + 8 \sum_{k=\pi(t)}^{t-1} \left( \delta_{1,k} + \delta_{2,k} \frac{\|\mathbf{z}_{k+1/2} - \mathbf{z}^*\|}{\lambda_k} \right) \overset{(26)}{\leq} 2R^2 + \frac{8R^2}{5} \leq 4R^2.$$

Thus, we have

$$\|\mathbf{z}_{t+1/2} - \mathbf{z}^*\|^2 \leq 2\|\mathbf{z}_{t+1/2} - \mathbf{z}_t\|^2 + 2\|\mathbf{z}_t - \mathbf{z}^*\|^2 \leq 9R^2. \qquad (28)$$

Summing up the batch of (25), for all $\mathbf{z} \in \mathbb{B}(\mathbf{z}^*, 3\sqrt{2}R)$, we have

$$\sum_{t=0}^{T-1} \frac{1}{\lambda_t} \langle \mathbf{F}(\mathbf{z}_{t+1/2}), \mathbf{z}_{t+1/2} - \mathbf{z} \rangle + \sum_{t=0}^{T-1} \frac{1}{8} \|\mathbf{z}_{t+1/2} - \mathbf{z}_t\|^2$$

$$\leq \frac{1}{2}(\|\mathbf{z}_0 - \mathbf{z}\|^2 - \|\mathbf{z}_T - \mathbf{z}\|^2) + \sum_{t=0}^{T-1} \left( \delta_{1,t} + \frac{\delta_{t,2}\|\mathbf{z}_{t+1/2} - \mathbf{z}\|}{\lambda_t} \right) \tag{29}$$

$$\leq \frac{R^2}{20} + \frac{R^2}{10} + \frac{TR^2(\|\mathbf{z}_{t+1/2} - \mathbf{z}^*\| + \|\mathbf{z} - \mathbf{z}^*\|)}{10T(\|\mathbf{z}_{t+1/2} - \mathbf{z}_0\| + R)} \leq 2R^2.$$

We then bound the regularization term $\sum_{t=0}^{T-1} \lambda_t^2$ can be bounded by

$$\sum_{t=0}^{T-1} \lambda_t^2 \leq \sum_{t=0}^{T-1} 36(3M^2\|\mathbf{z}_{t+1/2} - \mathbf{z}_t\|^2 + 3\delta_{\mathbf{H}}^2 + 3\delta_{1,t})$$

$$\leq 432M^2R^2 + \sum_{t=0}^{T-1}\left( \frac{3R^2}{T} + \frac{3R^2}{10T} \right) \leq (432M^2 + 4)R^2.$$

Finally, we have

$$\mathrm{Gap}(\bar{\mathbf{z}}_T; 3R) \leq \max_{\|\mathbf{z} - \mathbf{z}^*\| \leq 3\sqrt{2}R} \langle \mathbf{F}(\mathbf{z}), \bar{\mathbf{z}}_T - \mathbf{z} \rangle = \max_{\|\mathbf{z} - \mathbf{z}^*\| \leq 3\sqrt{2}R} \frac{1}{\sum_{t=0}^{T-1} 1/\lambda_t} \sum_{t=0}^{T-1} \frac{1}{\lambda_t} \langle \mathbf{F}(\mathbf{z}), \mathbf{z}_{t+1/2} - \mathbf{z} \rangle$$

$$\leq \max_{\|\mathbf{z} - \mathbf{z}^*\| \leq 3\sqrt{2}R} \frac{1}{\sum_{t=0}^{T-1} 1/\lambda_t} \sum_{t=0}^{T-1} \frac{1}{\lambda_t} \langle \mathbf{F}(\mathbf{z}_{t+1/2}), \mathbf{z}_{t+1/2} - \mathbf{z} \rangle \overset{(29)}{\leq} \frac{2R^2}{\sum_{t=0}^{T-1} 1/\lambda_t}$$

$$\leq \frac{2\sqrt{432M^2 + 4}R^3}{T^{3/2}},$$

where the last inequality is due to Lemma A.1. $\qquad\square$

## C.4 The Proof of Theorem 4.5

*Proof.* According to Lemmas 3.2 and 3.5, we know that $\mathbf{g}_t$, $\mathbf{H}$, $\mathbf{v}_t$ can be constructed within $\tilde{\mathcal{O}}(1)$, $\tilde{\mathcal{O}}(d)$, and $\tilde{\mathcal{O}}(1)$ quantum function evaluation oracle, respectively and

$$\|\mathbf{g}_t - \mathbf{F}(\mathbf{z}_t)\| \leq \|\mathbf{J}\|\|\tilde{\mathbf{g}}_t - \nabla f(\mathbf{z}_t)\| \leq 1500d\sqrt{L_1 \epsilon_{1,t}} \leq \frac{R^2}{10T},$$

$$\|\mathbf{H} - \nabla\mathbf{F}(\mathbf{z}_{\pi(t)})\| \leq \|\mathbf{J}\|\|\tilde{\mathbf{H}} - \nabla^2 f(\mathbf{z}_{\pi(t)})\| \leq 10\sqrt{15}d^2 L_1^{1/4}L_2^{1/2}\epsilon_{\mathbf{H}}^{1/4} \leq \frac{R}{\sqrt{T}},$$

and

$$\|\mathbf{v}_t - \mathbf{F}(\mathbf{z}_{t+1/2})\| \leq \|\mathbf{J}\|\|\tilde{\mathbf{v}}_t - \nabla f(\mathbf{z}_{t+1/2})\|$$

$$\leq 1500d\sqrt{L_1 \epsilon_{1,t}} \leq \min\left\{ \frac{\lambda_t R^2}{10T(\|\mathbf{z}_{t+1/2} - \mathbf{z}_0\| + R)}, \frac{R^2}{2T} \right\}$$

hold with probability at least $\left(1 - \frac{\delta}{T}\right)$.

Let $\delta_{1,t} = \frac{R^2}{10T}$, $\delta_{\mathbf{H},t} = \frac{R}{\sqrt{T}}$, and $\delta_{2,t} = \min\left\{ \frac{\lambda_t R^2}{10T(\|\mathbf{z}_{t+1/2} - \mathbf{z}_0\| + R)}, \frac{\delta_{1,t}}{2} \right\}$, we know that the condition of Lemma 4.4 holds with probability at least $(1 - \delta)$. Thus, the output $\bar{\mathbf{z}}_T$ of Algorithm 4 satisfies that

$$\mathrm{Gap}(\bar{\mathbf{z}}_T; \sqrt{3}R) \leq \frac{2\sqrt{432M^2 + 4}R^3}{T^{3/2}} \leq \epsilon,$$

with probability at least $(1 - \delta)$. The total number of queries to the quantum evaluation oracle can be bounded by

$$\#\mathrm{Query} = T \cdot \tilde{\mathcal{O}}(1) + \left( \frac{T}{m} + 1 \right) \tilde{\mathcal{O}}(d) = \tilde{\mathcal{O}}\left( d + L_2^{2/3}d^{2/3}\epsilon^{-2/3} \right).$$

$\qquad\square$

## C.5 The Proof of Lemma 4.7

*Proof.* We can obtain a good approximation of $\mathbf{F}(\mathbf{z}_{i+1/2}^{(s)})$, $\mathbf{F}(\mathbf{z}_i^{(s)})$, and $\nabla\mathbf{F}(\mathbf{z}_{\pi(t)}^{(s)})$ with probability at least $(1 - \delta/S)$ for all $i \in [T-1]$. Recalling the proof of Lemma 4.4 in Appendix C.3, from (25), we have

$$\sum_{i=\pi(t)}^{t-1} \left( \frac{1}{\lambda_i^{(s)}} \langle \mathbf{F}(\mathbf{z}_{i+1/2}^{(s)}), \mathbf{z}_{i+1/2}^{(s)} - \mathbf{z} \rangle + \frac{1}{8}\|\mathbf{z}_{i+1/2}^{(s)} - \mathbf{z}_i^{(s)}\|^2 \right)$$

$$\leq \frac{1}{2} \left( \|\mathbf{z}_{\pi(t)}^{(s)} - \mathbf{z}\|^2 - \|\mathbf{z}_t^{(s)} - \mathbf{z}\|^2 \right) + \sum_{i=\pi(t)}^{t-1} \left( \frac{3}{16}\delta_{1,i}^{(s)} + \frac{\delta_{2,i}^{(s)}}{\lambda_i^{(s)}}\|\mathbf{z}_{i+1/2}^{(s)} - \mathbf{z}\| \right) \tag{30}$$

holds with probability at least $(1 - \delta/S)$. Assumption 4.6 means that

$$\langle \mathbf{F}(\mathbf{z}) - \mathbf{F}(\mathbf{z}'), \mathbf{z} - \mathbf{z}' \rangle \geq \mu\|\mathbf{z} - \mathbf{z}'\|^2. \tag{31}$$

Let $\mathbf{z} = \mathbf{z}^*$ in (30), we have

$$\sum_{i=\pi(t)}^{t-1} \left( \frac{\mu}{\lambda_i^{(s)}} \|\mathbf{z}_{i+1/2}^{(s)} - \mathbf{z}^*\|^2 + \frac{1}{8}\|\mathbf{z}_{i+1/2}^{(s)} - \mathbf{z}_i^{(s)}\|^2 \right)$$

$$\leq \frac{1}{2}(\|\mathbf{z}_{\pi(t)}^{(s)} - \mathbf{z}^*\|^2 - \|\mathbf{z}_t^{(s)} - \mathbf{z}\|^2) + \sum_{i=\pi(t)}^{t-1} \left( \frac{3}{16}\delta_{1,i}^{(s)} + \frac{\delta_{2,i}^{(s)}}{\lambda_i^{(s)}}\|\mathbf{z}_{i+1/2}^{(s)} - \mathbf{z}\| \right)$$

Summing up above inequality from $i = 0$ to $i = T - 1$, we have

$$\sum_{i=0}^{T-1} \left( \frac{\mu}{\lambda_i^{(s)}} \|\mathbf{z}_{i+1/2}^{(s)} - \mathbf{z}^*\|^2 + \frac{1}{8}\|\mathbf{z}_{i+1/2}^{(s)} - \mathbf{z}_i^{(s)}\|^2 \right) \leq \frac{1}{2}\|\mathbf{z}_0^{(s)} - \mathbf{z}^*\|^2 + \sum_{i=0}^{T-1} \left( \frac{3}{16}\delta_{1,t}^{(s)} + \frac{\delta_{2,i}^{(s)}}{\lambda_i^{(s)}}\|\mathbf{z}_{i+1/2}^{(s)} - \mathbf{z}^*\| \right).$$

The choice of $\delta_{1,t}^{(s)}$ and $\delta_{2,t}^{(s)}$ guarantees that

$$\sum_{i=0}^{T-1} \left( \frac{3}{16}\delta_{1,t}^{(s)} + \frac{\delta_{2,i}^{(s)}}{\lambda_i^{(s)}}\|\mathbf{z}_{i+1/2}^{(s)} - \mathbf{z}^*\| \right) \leq \frac{1}{4}\|\mathbf{z}_0^{(s)} - \mathbf{z}^*\|^2.$$

Thus we have

$$\sum_{i=0}^{T-1} \|\mathbf{z}_{i+1/2}^{(s)} - \mathbf{z}_i^{(s)}\|^2 \leq 6\|\mathbf{z}_0^{(s)} - \mathbf{z}^*\|^2.$$

Since $\mathbf{z}^{(s+1)} = \bar{\mathbf{z}}_T^{(s)} = \frac{\sum_{i=0}^{T-1} \frac{\mathbf{z}_i^{(s)}}{\lambda_i^{(s)}}}{\sum_{i=0}^{T-1} \frac{1}{\lambda_i^{(s)}}}$ and $\mathbf{z}^{(s)} = \mathbf{z}_0^{(s)}$, using the convexity of $\|\cdot\|^2$ and by Jensen's inequality, we have

$$\|\mathbf{z}^{(s+1)} - \mathbf{z}^*\|^2 \leq \frac{3}{4\mu \sum_{i=0}^{T-1} \frac{1}{\lambda_i^{(s)}}}\|\mathbf{z}^{(s)} - \mathbf{z}^*\|^2. \tag{32}$$

We then bound the term $\sum_{i=0}^{T-1}(\lambda_i^{(s)})^2$ by

$$\sum_{t=0}^{T-1}(\lambda_t^{(s)})^2 \leq \sum_{t=0}^{T-1} 36(3M^2\|\mathbf{z}_{t+1/2}^{(s)} - \mathbf{z}_t^{(s)}\|^2 + 3(\delta_{\mathbf{H}}^{(s)})^2 + 3\delta_{1,t}^{(s)}) \leq \left(648M^2 + 6\right)\|\mathbf{z}^{(s)} - \mathbf{z}^*\|^2.$$

Since

$$T = \left\lceil \frac{(100M + 12)^{2/3}\|\mathbf{z}^{(0)} - \mathbf{z}^*\|^{2/3}}{\mu^{2/3}} \right\rceil,$$

it enough to guarantee the linear decent on $\|\mathbf{z}^{(s)} - \mathbf{z}^*\|$ such that

$$\|\mathbf{z}^{(s+1)} - \mathbf{z}^*\|^2 \leq \frac{3}{4\mu \sum_{i=0}^{T-1} \frac{1}{\lambda_i^{(s)}}}\|\mathbf{z}^{(s)} - \mathbf{z}^*\|^2 \leq \frac{3\sqrt{648M^2 + 6}\|\mathbf{z}^{(s)} - \mathbf{z}^*\|^3}{4\mu T^{3/2}} \leq \frac{1}{2}\|\mathbf{z}^{(s)} - \mathbf{z}^*\|^2.$$

$\square$

## C.6 The Proof of Theorem 4.8

*Proof.* Using Lemma 4.7, we have that $\|\mathbf{z}^{(s+1)} - \mathbf{z}^*\|^2 \leq \frac{1}{2}\|\mathbf{z}^{(s)} - \mathbf{z}^*\|^2$ holds for all $s \in [S-1]$ with probability at least $1 - \delta$. Then we have $\|\mathbf{z}^{(S)} - \mathbf{z}^*\|^2 \leq \left(\frac{1}{2}\right)^S \|\mathbf{z}^{(0)} - \mathbf{z}^*\|^2 \leq \epsilon$. For each call of the subroutine HAQZO$^+$, the query complexity can be bounded by

$$T \cdot \tilde{\mathcal{O}}(1) + \frac{T}{m} \cdot \tilde{\mathcal{O}}(d) = \tilde{\mathcal{O}}\left(d + \frac{d^{2/3}L_2^{2/3}}{\mu^{2/3}}\right).$$

Thus, the total query complexity can be bounded by

$$\#\text{Query} = T \cdot S = \left(T \cdot \tilde{\mathcal{O}}(1) + \left(\frac{T}{m} + 1\right)\tilde{\mathcal{O}}(d)\right)\log(\epsilon^{-1}) = \tilde{\mathcal{O}}\left(d + L_2^{2/3}d^{2/3}\mu^{-2/3}\right).$$

$\square$

# D The Proof of Section 5

We first present some useful results for one step of lazy CRN [14, 13]:

$$\mathbf{z}_{t+1} = \underset{\mathbf{z} \in \mathbb{R}^d}{\arg\min}\left\{\langle \mathbf{g}_t, \mathbf{z} - \mathbf{z}_t\rangle + \frac{1}{2}\langle \mathbf{H}_{\pi(t)}(\mathbf{z} - \mathbf{z}_t), \mathbf{z} - \mathbf{z}_t\rangle + \frac{M}{6}\|\mathbf{z} - \mathbf{z}_t\|^3\right\}, \tag{33}$$

where $\mathbf{g}_t$ and $\mathbf{H}_{\pi(t)}$ are good estimations to $\nabla f(\mathbf{z}_t)$ and $\nabla^2 f(\mathbf{z}_{\pi(t)})$, respectively, such that

$$\|\mathbf{g}_t - \nabla f(\mathbf{z}_t)\|_2 \leq \delta_{\mathbf{g}} \quad \text{and} \quad \left\|\mathbf{H}_{\pi(t)} - \nabla^2 f(\mathbf{z}_{\pi(t)})\right\| \leq \delta_{\mathbf{H}}. \tag{34}$$

**Lemma D.1** (Theorem 2.4 [13]). *Consider the cubic regularization step in* (33) *where* $\mathbf{g}_t$ *and* $\mathbf{H}_{\pi(t)}$ *satisfy* (34), *then it holds that*

$$\begin{aligned}
f(\mathbf{z}_t) - f(\mathbf{z}_{t+1}) &\geq \frac{1}{192M^{1/2}}\|\nabla f(\mathbf{z}_{t+1})\|^{3/2} \\
&+ \left(\frac{M}{48}\|\mathbf{z}_{t+1} - \mathbf{z}_t\|^3 - \frac{171L_2^3}{M^2}\|\mathbf{z}_t - \mathbf{z}_{\pi(t)}\|^3 - \frac{171\delta_{\mathbf{H}}^3}{M^2} - \frac{3\delta_{\mathbf{g}}^{3/2}}{M^{1/2}}\right).
\end{aligned} \tag{35}$$

Then we know that by properly choosing $\delta_{\mathbf{g}}$, $\delta_{\mathbf{H}}$, the CRN step can make the gradient small with a rate of $\mathcal{O}(T^{-3/2})$.

**Lemma D.2.** *Under Assumptions 2.2, let* $\bar{\mathbf{z}}$ *be uniformly chosen from* $\{\mathbf{z}_i\}_{i=1}^T$, *generated by* (33), *then it holds that*

$$\mathbb{E}\left[\|\nabla f(\bar{\mathbf{z}})\|^{3/2}\right] \leq \frac{192\sqrt{M}(f(\mathbf{z}_0) - f^*)}{T} + \left(576\delta_{\mathbf{g}}^{3/2} + \frac{200^2\delta_{\mathbf{H}}^3}{M^{3/2}}\right)$$

*Proof.* Summing up (35) from $k = \pi(t)$ to $t$, then it holds that

$$f(\mathbf{z}_{\pi(t)}) - f(\mathbf{z}_t)$$

$$\geq \frac{1}{192 M^{1/2}} \sum_{k=\pi(t)}^{t-1} \|\nabla f(\mathbf{z}_{k+1})\|^{3/2}$$

$$+ \sum_{k=\pi(t)}^{t-1} \left( \frac{M}{48} \|\mathbf{z}_{k+1} - \mathbf{z}_k\|^3 - \frac{171 L_2^3}{M^2} \|\mathbf{z}_k - \mathbf{z}_{\pi(t)}\|^3 \right) - \sum_{k=\pi(t)}^{t-1} \left( \frac{171 \delta_{\mathbf{H}}^3}{M^2} + \frac{3 \delta_{\mathbf{g}}^{3/2}}{M^{1/2}} \right)$$

$$\geq \frac{1}{192 M^{1/2}} \sum_{k=\pi(t)}^{t-1} \|\nabla f(\mathbf{z}_{k+1})\|^{3/2}$$

$$+ \sum_{k=\pi(t)}^{t-1} \left( \frac{M r_k^3}{48} - \frac{171 L_2^3 (\sum_{j=\pi(t)}^{k-1} r_j)^3}{M^2} \right) - \sum_{k=\pi(t)}^{t-1} \left( \frac{171 \delta_{\mathbf{H}}^3}{M^2} + \frac{3 \delta_{\mathbf{g}}^{3/2}}{M^{1/2}} \right) \qquad (36)$$

$$\geq \frac{1}{192 M^{1/2}} \sum_{k=\pi(t)}^{t-1} \|\nabla f(\mathbf{z}_{k+1})\|^{3/2}$$

$$+ \sum_{k=\pi(t)}^{t-1} \left( \frac{M}{48(t - \pi(t))^3} - \frac{171 L_2^3}{M^2} \right) \left( \sum_{j=\pi(t)}^{k-1} r_j \right)^3 - \sum_{k=\pi(t)}^{t-1} \left( \frac{171 \delta_{\mathbf{H}}^3}{M^2} + \frac{3 \delta_{\mathbf{g}}^{3/2}}{M^{1/2}} \right)$$

$$\geq \frac{1}{192 M^{1/2}} \sum_{k=\pi(t)}^{t-1} \|\nabla f(\mathbf{z}_{k+1})\|^{3/2} - \sum_{k=\pi(t)}^{t-1} \left( \frac{171 \delta_{\mathbf{H}}^3}{M^2} + \frac{3 \delta_{\mathbf{g}}^{3/2}}{M^{1/2}} \right),$$

where the last inequality is due to Lemma A.3 and $M = 30 m L_2$

$$\frac{M}{48(t - \pi(t))^3} - \frac{171 L_2^3}{M^2} \geq \frac{M}{144 m^3} - \frac{171 L_2^3}{M^2} \geq 0.$$

Summing up (36) from 0 to $T - 1$, we have

$$f(\mathbf{z}_0) - f^* \geq f(\mathbf{z}_0) - f(\mathbf{z}_T) \geq \frac{1}{192 M^{1/2}} \sum_{t=0}^{T-1} \|\nabla f(\mathbf{z}_{t+1})\|^{3/2} - T \left( \frac{171 \delta_{\mathbf{H}}^3}{M^2} + \frac{3 \delta_{\mathbf{g}}^{3/2}}{M^{1/2}} \right),$$

and

$$\mathbb{E}\left[ \|\nabla f(\bar{\mathbf{z}})\|^{3/2} \right] = \frac{1}{T} \sum_{t=0}^{T-1} \|\nabla f(\mathbf{z}_{t+1})\|^{3/2} \leq \frac{192 \sqrt{M}(f(\mathbf{z}_0) - f^*)}{T} + \left( 576 \delta_{\mathbf{g}}^{3/2} + \frac{200^2 \delta_{\mathbf{H}}^3}{M^{3/2}} \right).$$

$\square$

Now we are ready to prove Theorem 5.2.

### D.1 The Proof of Theorem 5.2

*Proof.* The choice of $\epsilon_{\mathbf{g}}$ and $\epsilon_{\mathbf{H}}$ means that

$$\|\mathbf{g}_t - \nabla f(\mathbf{z}_t)\| \leq \frac{\epsilon^{-1}}{1728^{2/3}} := \delta_{\mathbf{g}}$$

$$\|\mathbf{H}_{\pi(t)} - \nabla^2 f(\mathbf{z}_{\pi(t)})\| \leq \frac{M^{1/2} \epsilon^{-1/2}}{30} := \delta_{\mathbf{H}}$$

hold with probability at least $(1 - \delta)$ for all $t \in [T]$. Using Lemma D.2, we have

$$\mathbb{E}\left[ \|\nabla f(\mathbf{z}_{\text{out}})\|^{3/2} \right] \leq \epsilon^{-3/2},$$

due to the convexity of $x^{3/2}$, we have

$$\mathbb{E}\left[\|\nabla f(\mathbf{z}_{\text{out}})\|\right] \leq \left(\mathbb{E}\left[\|\nabla f(\mathbf{z}_{\text{out}})\|^{3/2}\right]\right)^{2/3} \leq \epsilon^{-1}.$$

We require using $\tilde{\mathcal{O}}(1)$ query complexity of function evaluation oracle to construct the gradient estimator for all $T$ iterations and using $\tilde{\mathcal{O}}(d)$ query complexity to construct the Hessian estimator for $\frac{T}{m} + 1$ iterations at the snapshot point. Thus, the total query complexity of Algorithm 6 can be bounded by

$$\#\text{Query} = T \cdot \tilde{\mathcal{O}}(1) + \left(\frac{T}{m} + d\right)\tilde{\mathcal{O}}(d) = \tilde{\mathcal{O}}\left(d + d^{1/2}L_2^{1/2}(f(\mathbf{z}_0) - f(\mathbf{z}^*))\epsilon^{-3/2}\right).$$

$\square$

