# OpenReview forum: "Quantum Speedups for Minimax Optimization and Beyond"
_NeurIPS.cc/2025/Conference — NeurIPS 2025 poster_

### Official Review · Reviewer_iZ6K · 2025-06-02

**Clarity:** 2
**Significance:** 3
**Originality:** 3
**Rating:** 5
**Confidence:** 4

**Summary:**

The paper tackles convex–concave minimax optimization with only zeroth-order (function-value) access by introducing Hessian-aware quantum zeroth-order algorithms that use fast quantum estimators requiring $\tilde O(1)$ gradient and $\tilde{O}(d)$ Hessian queries. The main scheme, HAQZO, achieves query complexity $\tilde{O}(d\epsilon^{-2/3})$, while a double-loop variant, HAQZO+, improves this to $\tilde{O}(d^{2/3}\epsilon^{-2/3})$, surpassing the classical $\tilde{O}(d\epsilon^{-1})$ rate. For $\mu$-strongly-convex–strongly-concave games, a restarted method reaches $\tilde{O}(d^{2/3}(L_2/\mu)^{2/3})$ queries. Extending to non-convex problems, a Quantum Cubic-Regularised Newton Method (QCNM) finds $\epsilon$-stationary points in $\tilde{O}(d^{1/2}\epsilon^{-3/2})$ oracle calls. These complexities establish quantum speedups for gradient-free optimization convex, strongly-convex, and non-convex settings.

**Questions:**

I hope the authors can respond and address my minor comment in the Strengths And Weaknesses part.

**Ethical Concerns:**

["NO or VERY MINOR ethics concerns only"]

**Final Justification:**

This is a technically solid work providing quantum algorithms with provably better performance for minimax optimization over best-known classical counterparts given queries to only zero-th order queries to the function. As far as I know, this is the first work that provides quantum algorithms that outperform classical algorithms in all settings of minimax optimization given queries to either function value or gradients. In addition, the result is not a direct application of the well-known quantum gradient estimation algorithm. Instead, improved quantum Hessian-vector estimator and Hessian estimator are proposed with better performance than the best-known classical counterparts. Besides improving upon Jordan's algorithm to obtain a quantum Hessian estimator, I also find the part of inserting the lazy Hessian technique from classical literature to reduce the $d$ dependence very interesting and technically involved. Therefore, I am happy to vote for acceptance of this paper at NeurIPS 2025.

**Limitations:**

Yes

**Quality:**

3

**Strengths And Weaknesses:**

From my perspective, this is a technically solid work providing quantum algorithms with provably better performance for minimax optimization over best-known classical counterparts given queries to only zero-th order queries to the function. As far as I know, this is the first work that provides quantum algorithms that outperform classical algorithms in all settings of minimax optimization given queries to either function value or gradients. In addition, the result is not a direct application of the well-known quantum gradient estimation algorithm. Instead, improved quantum Hessian-vector estimator and Hessian estimator are proposed with better performance than the best-known classical counterparts. Therefore, I am happy to vote for acceptance of this paper at NeurIPS 2025.

The weakness of this paper, if I have to provide one, is that this work provides no understanding of whether the query complexity achieved by HAQZO+ or restart-HAOZO+ is optimal in certain parameter regimes. This is also mentioned by the authors in the conclusion. However, I believe the contribution of this work by proposing the quantum algorithm with a provable quantum advantage is already significant enough.

Besides, a minor comment is that I believe the overall presentation of the work can be significantly improved by adding a more detailed literature review regarding classical minimax optimization in different settings, including (strong) convex-(strong) concave functions, non-convex-(strong) concave functions, zeroth-order queries, first-order queries, etc.

---

> ### Author Rebuttal · Authors · 2025-07-29
>
> Thanks for your helpful comments and recognition on our work.
>
> ## To weakness 1:
>
> For the convex-concave minimax problems, the lower bound for zeroth-order algorithms are even unknown for classical algorithms.
>
> For the non-convex minimization problems, some negative results claim that there is no quantum speed-up for higher-order methods when the dimension is large, i.e. the best dependency is $\epsilon^{-2/3}$ even with second-order oracles. The lower bound for this is also unknown even for classical algorithms.
>
> We will discuss this in our revision and leave it as an interesting future work.
>
> ## To weakness 2:
>
> We will add a more detailed literature review regarding classical minimax optimization in the revision.

---

> > ### Comment · Reviewer_iZ6K · 2025-08-01
> >
> > I thank the authors for the kind explanation. My previously mentioned weaknesses are minor ones and would not affect my overall judgment about this work. In particular, figuring out the lower bound should be an independent theoretical problem and can be very hard in most cases. Besides improving upon Jordan's algorithm to obtain a quantum Hessian estimator, I also find the part of inserting the lazy Hessian technique from classical literature to reduce the $d$ dependence very interesting and technically involved. I would like to keep my score unchanged.

---

> > > ### Author Response · Authors · 2025-08-01
> > >
> > > Thanks for your recognition and support to our work.

---

### Official Review · Reviewer_eEJb · 2025-06-21

**Clarity:** 3
**Significance:** 3
**Originality:** 4
**Rating:** 5
**Confidence:** 3

**Summary:**

The authors provide a method for solving $\epsilon$ min-max problems using a zeroth order quantum method that approximates Hessians.  Their algorithm claims a lower query complexity than previous work, improving upon previously known classical methods by a factor of $d^{1/3} \epsilon^{1/3}$ for convex-concave problems, and $d^{1/3} \mu^{1/3}$ for strongly convex-concave problems, where $d$  is the dimension of the domain, $\epsilon$ is the error in the saddle point estimate, and $\mu$ is the coefficient determining the strength of the strongly concave-convex problems.  The method relies on an oracle which computes function evaluations for a function $f$, which is convex in the first variable and concave in the second variable, as well as Lipschitz continuity in the 0th, 1st, and 2nd derivatives.

**Questions:**

How well does this method generalize to functions with degenerate saddle points?

How will the oracle implementation effect the performance of this algorithm?  It strikes me that the algorithm scales with $d^{2/3}$, but just loading the problem would take at least $d$, which could limit your performance.  Is this a limitation?

$x$ and $y$ are said to be real valued, but it is not obvious to me how you would embed real-valued, rather than simply natural valued inputs (i.e. a linear combination over the computational basis).  Does this restriction on your input $x$ and $y$ values pose a problem for your algorithm or is there an amplitude embedding, etc. that avoids this issue (or is it a non-issue)?

**Ethical Concerns:**

["NO or VERY MINOR ethics concerns only"]

**Final Justification:**

Upon reading their responses and giving another look through of the paper, I improved their clarity score.  The material is just dense - it is not poorly written.  My overall score remains at accept.  This is a solid paper in my mind.

**Limitations:**

It is unclear how generally applicable this method is to real world problem instances.  They claim that these types of problems are important for "fairness aware learning, AUC maximization, robust optimization, game theory, and reinforcement learning," but it is unclear to me what problem scales of these problems become relevant.  This is important, since as was shown in M. Beverland, 2022 (https://arxiv.org/abs/2211.07629), scaling advantages that are sub-quadratic are unlikely to be practically realizable on currently considered quantum hardware.  Again, this is a theory paper, so this limitation is not particularly major in the context of this work, but it is a practical limitation nonetheless.

**Quality:**

3

**Strengths And Weaknesses:**

The paper is very technically solid, and the appendices go into strong detail about many of the derivations.  They do a good job of listing the core assumptions of their method as well as providing very detailed appendices outlining the derivation of their key lemmas.  That said, the overall readability of the main text is somewhat difficult to follow - most likely due to the page limit.  I would have liked it if they referred readers to the appendices for proofs of their key points in the main text to make it easier to follow where some of the bounds were coming from.  It was also somewhat unclear how viable this method would be to implement on a fault tolerant computer, as hidden overheads in the oracle embedding or mapping onto hardware for general functions is not clear.  That said, this was a theory paper, and those types of concerns are beyond the scope of this work.  There was nothing overall wrong technically with their paper, as far far as I could tell, and it provides a good theoretical foundation for further work in the future.  Therefore I believe that the paper should be accepted.

---

> ### Author Rebuttal · Authors · 2025-07-29
>
> Thanks for your helpful comments and recognition in our work.
>
> ## To question 1:
>
> This method can indeed solve the minimax problems with degenerate saddle point. Both Theorem 4.2 and Theorem 4.4 require only convex-concave assumptions on the objective, which allows the existence of degenerate saddle point.
>
> ## To question 2:
>
> Thanks for pointing this out, loading this problem takes $\mathcal{O}(d)$, but only at a very few iterations. The total query complexity is $\mathcal{O}(d+d^{2/3}\epsilon^{-2/3})$ where the $\mathcal{O}(d)$ factor is only an addition term and can be omitted when $d$ is small $(d\leq \epsilon^{-2})$.
>
> ## To question 3:
> We do not use amplitude encoding for the real-valued inputs. Instead, we encode these values into the computational basis using a fixed-precision binary representation, analogous to how floating-point numbers are handled in classical computing.
>
> This approach is standard in the analysis of quantum algorithms, just as it is for classical algorithms designed for classical electronic computers. Classical computers also operate on finite-precision representations of real numbers. Notice that, we assume that the value oracle of $f$ has an error $\epsilon_0$, which is intended to account for such sources of imprecision, including the truncation errors that arise from this finite-precision encoding.
>
> For the sake of clarity and notational convenience, we present our analysis assuming the optimization occurs over the real numbers. This is also a common practice in the quantum optimization literature, with the implicit understanding that any practical implementation would rely on a suitable finite-precision encoding. This simplification does not pose a problem for the validity of our algorithm.

---

> > ### Comment · Reviewer_eEJb · 2025-08-05
> > **Thank you for your response**
> >
> > This was a well thought out and considered response, thank you.  The point about fixed point representations makes good sense to me, so thank you again.
> >
> > Upon a brief re-consideration, I decided to increment your clarity score by one, as I don't think that the paper was unclear, I think the material was just dense, which was out of the authors control.
> >
> > My final score remained unchanged, as I continue to believe that this paper should be accepted.

---

> > > ### Author Response · Authors · 2025-08-05
> > >
> > > Thanks for your recognition and support to our work!

---

### Official Review · Reviewer_77Pf · 2025-06-30

**Clarity:** 2
**Significance:** 2
**Originality:** 2
**Rating:** 4
**Confidence:** 3

**Summary:**

This work studies quantum algorithms for zeroth order optimization problems, and specifically, for the minimax problem. The authors introduce a class of Hessian-aware quantum zeroth-order methods that achieve speedups over classical methods in terms of query complexity to the function value oracle.

**Questions:**

Typos:
- Line 136: exits -> exists;

The writing could be improved. Many theorem statements has grammar mistakes.

Could the authors give more explanations about the novelty of their paper? As far as I can see, the quantum Hessian estimation is a direct application of the quantum gradient estimation method. Whether the method is optimal or not still needs discussion.

**Ethical Concerns:**

["NO or VERY MINOR ethics concerns only"]

**Final Justification:**

Issues solved:
- I was previously concerned about the main technical novelties of the paper, as the quantum gradient estimation and Hessian estimation has already been discussed in previous literatures like [Jor05, GAW19, ZS24]. After the rebuttal, and especially seeing the discussions between Reviewer mnz2 and authors, I think that a main novelty lies in developing a new ``inexact'' classical gradient-based optimization algorithm with better performance, and the classical framework cooperates well with quantum subroutines.

- Another point that has been clarified is the query complexity part. The authors has claimed that $\mathcal{O}(d+d^{1/2}\epsilon^{-3/2})=\mathcal{O}(d^{1/2}\epsilon^{-3/2})$. I think the additive term $\mathcal{O}(d)$ should not be omitted in the main theorem.

Issues remain unresolved:
- Computational complexity of their algorithms, which is also considered by Reviewer iyXU. Although the authors have provided certain examples about why query complexity could be a dominant term, I am still unclear about the overall time complexity of their algorithms, and the resources needed (like the number of qubits, number of gates, and depth of the circuit, et.c.). I think explicilty showing the computational complexity of the algorithms is rather important for quantum algorithms, and will help us better understand when quantum could provide end-to-end speedups.

- Novelty for quantum algorithm design. Although the authors have made progress in designing quantum gradient and Hessian estimation, I feel their work in this aspect incremental. It may not be so inspiring for quantum people to design new quantum algorithms based on their work.

**Limitations:**

The work fails to give lower bounds for query complexity of quantum algorithms in the problem setting, leaving the optimality of their algorithms as an open question.
The method of estimating the Hessian matrix seems to be a direct application of quantum gradient estimation, which might not be so novel.

**Paper Formatting Concerns:**

No major formatting issues in the paper.

**Quality:**

2

**Strengths And Weaknesses:**

Strength:

- The paper has proposed new quantum zeroth order optimization algorithms, with provable speedups in query complexity than classical algorithms.
- The paper has given new quantum algorithms for estimating the Hessian matrix.

Weakness:

- The acceleration in query complexity of the methods are modest, less than quadratic.
- In the quantum gradient and Hessian estimation, the gate complexity is similar to the one in the classical case. Therefore, I guess the reason why focusing on the query complexity should be clearly explained with examples.
- The work fails to give lower bounds for query complexity of quantum algorithms in the problem setting.

---

> ### Author Rebuttal · Authors · 2025-07-29
>
> Thanks for your helpful comments.
>
> ## To weakness 1:
>
> For convex-concave minimax problems, the state-of-the-art classical algorithm requires  $\mathcal{O}(d\epsilon^{-1})$  queries to find an $\epsilon$-saddle point, exhibiting a complexity that is linear in both $d$ and $\epsilon^{-1}$. In contrast, our algorithm (HAQZO+, Algorithm 4) can solve this problem with a query complexity that is **sublinear** in both $d$ and $\epsilon^{-1}$.
>
> For non-convex minimization problems, our algorithm (QCNM, Algorithm 6) with query complexity of $\mathcal{O}(d+d^{1/2}\epsilon^{-3/2})$demonstrates  **at least a quadratic speedup on the dependency of dimension** upon the prior zeroth-order cubic-regularized-Newton methods $\mathcal{O}(d^2+d^{3/2}\epsilon^{-3/2})$.
>
> ## To weakness 2:
>
> We consider the case where the query complexity dominates the computation complexity of the algorithm, which is a natural setting in both classical and quantum optimization. For example, considering the generalized linear model such that  $f(x)=h(A^{\top}x)$ where $A\in R^{d\times n}$, the complexity of one query is at least $nd$, which can be much larger than the gate complexity if $n\gg d$. Our algorithm achieves meaningful quantum speedups under such a setting, typically when the dimension is not too large.
>
> ## To weakness 3:
>
> For the convex-concave minimax problems, the lower bound for zeroth-order algorithms are even unknown for classical algorithms.
>
> For the non-convex minimization problems, some negative results claim that there are no-quantum speed-up for higher-order methods when the dimension is large [A], i.e. the best dependency is $\epsilon^{-2/3}$ even for algorithms with second-order oracles. The lower bound for $d$ is also  unknown even for classicla algorithms.
>
> We will discuss this in our revision and leave it as an interesting future work.
>
> [A]. Quantum Lower Bounds for Finding Stationary Points of Nonconvex Functions. ICML, 2023.
>
> ## To Question 1 (typos and writing)：
>
> Thanks for pointing out this, we will modify them in the revision.
>
> ## To Question 2:
>
> We give explanations about the novelty in this paper below.
>
> From the quantum mechanism perspective, we give the first analytitcal results on the query complexity for estimating Hessian-vector products and Hessian with an arbitrary desired accuracy $\epsilon$, which are important oracles in quantum optimization.
>
> Besides the above, **we emphasize that another important contribution of our paper is that our algorithm is the first one to allow both inexactness of Hessian and gradient oracles to solve the minimax problems**, which is more general than the prior classical Hessian-aware algorithm frameworks [9, 27]. To achieve this, our algorithm requires novel design on the regularization term $\lambda_t$ and techniques to bound the errors from both quantum gradient and Hessian estimations.
>
> From the perspective of theoretical results, our algorithm demonstrates quantum speedup on general (strongly)-convex-(strongly)-concave minimax problems. Furthermore, the idea of accelerating classical algorithms by quantum Hessian estimation is new to quantum optimization, which is of independent research interest and can be used to solve other optimization problems. For example, the presented QCNM (Algorithm 6) is even faster than existing quantum algorithms when $d\leq \epsilon^{-1/4}$ for non-convex problems.
>
> ## To limitations:
>
> Please refer to our response to weakness 3 and question 2 respectively.

---

> > ### Comment · Reviewer_77Pf · 2025-08-03
> >
> > I thank the authors for their kind explanation, which has solved many of my concerns. One quick follow-up question regarding the complexity $\mathcal{O}(d+d^{1/2}\epsilon^{-3/2})$ you mentioned is not the same as the theorem 5.2 (or in the Section 1, table 3), which is  $\mathcal{O}(d^{1/2}L_2^{1/2}(f(z_0)-f^*)\epsilon^{-3/2})$, so is there anything missing in the text?

---

> > > ### Author Response · Authors · 2025-08-03
> > >
> > > Thanks for your follow-up question.
> > >
> > > The $\mathcal{O}(d)$ factor comes from the estimation of the full Hessian at the snapshot points. In our Theorem 5.1 (or in the Section 1, table 3), we omit the $\mathcal{O}(d)$ factor since it is only an additiion term in the query complexity and the dominated part is $\mathcal{O}(d^{1/2}\epsilon^{-3/2})$ when $d$ is not large, i.e. when $d\leq \epsilon^{-3}$, we have $\mathcal{O}(d+d^{1/2}\epsilon^{-3/2})=\mathcal{O}(d^{1/2}\epsilon^{-3/2})$.

---

> > > ### Author Response · Authors · 2025-08-04
> > >
> > > Dear Reviewer 77Pf,
> > >
> > > We would like to know if our response has addressed your follow-up question?
> > >
> > > Best regards,
> > >
> > > Authors

---

> ### Comment · Reviewer_77Pf · 2025-08-05
>
> Thanks for the clarification! I have no more questions. By the way, I think it will be clearer if the $O(d)$ additive term appears explicitly in the main text, maybe in the theorem or footnote.

---

> > ### Author Response · Authors · 2025-08-05
> >
> > Thanks for your suggestion, we will add the $\mathcal{O}(d)$ to the main text in revision. Please kindly reconsider your score if all of your concerns have been solved.

---

### Official Review · Reviewer_iyXU · 2025-07-01

**Clarity:** 3
**Significance:** 2
**Originality:** 3
**Rating:** 3
**Confidence:** 4

**Summary:**

This paper investigates quantum algorithms for convex-concave minimax optimization problems with access to a quantum zeroth-order oracle only. By combining a novel quantum Hessian vector estimator (based on the quantum gradient estimation subroutine) with existing classical (stochastic) second-order methods, such as Newton proximal extragradient, the proposed quantum algorithm achieves an improvement of $d^{1/3}\epsilon^{-1/3}$ over best-known classical counterparts. Application of the quantum Hessian estimator to non-convex optimization problems also achieves polynomial improvement in the query complexity compared to classical methods.

**Questions:**

Technical questions:
- How is the inexact cubic step in Algorithm 3 computed? If this step is performed purely classically, how can we expect any **genuine** quantum speedup, given that representing the Hessian as a classical array may already negate the quantum advantage gained from the Hessian estimation?
- The classical algorithms for convex-concave minimax optimization problems are already very good (with a provable query upper bound that is sublinear in both $d$ and $1/\epsilon$). In what parameter regime would we expect practical quantum advantage? Is it possible to have large (i.e., super-quadratic) quantum speedups?
- Lemma 4.1: Is the Gap() defined anywhere? I do not find it near the lemma, and it is not directly clear to me how this gap is defined.

Minor typos/formatting issues:
- Line 24: "When ... is Lipschitz continuous, second-order offers faster convergence rates." A word like "method" or "algorithm" is missing after "second-order".
- Line 29: the h(\cdot, \cdot) is not in the mathematical italic format.
- Line 57: There should be a space between the big-O notation and "complexity".
- Theorem 5.2: what does the $\Delta$ symbol mean in the in-line math equation: $f^* \Delta \min_{x\in \mathbb{R}^d}$?

**Ethical Concerns:**

["NO or VERY MINOR ethics concerns only"]

**Final Justification:**

See the comment for the authors.

**Limitations:**

– The main technical contribution (quantum Hessian estimation via a zeroth-order oracle) is relatively straightforward and appears to offer only limited benefit when integrated into a classical algorithm.
– The quantum speedups achieved are modest compared to classical baselines, and it remains unclear whether these would translate into a meaningful practical advantage once classical computation and post-processing overheads are accounted for.

**Paper Formatting Concerns:**

I have not noticed any major formatting issues in the paper.

**Quality:**

2

**Strengths And Weaknesses:**

Strengths:
- The Hessian-vector estimation subroutine is a novel technical contribution that can potentially accelerate other classical (second-order) optimization/minimax algorithms.
- The theoretical analysis appears complete and solid, with a comprehensive comparison with prior art in Tables 1-3.
- Extension to non-convex optimization problems is discussed, complementing the discussion on minimax optimization and showcasing the usefulness of the quantum Hessian estimation.

Weaknesses:
- While the Hessian-vector and Hessian estimation subroutines (Algorithms 1 & 2) appear original and have not been formally analyzed in previous work, they are quite straightforward extensions of the quantum gradient estimators (Lemma 3.2) relying on a standard 2-point estimation technique (e.g., finite difference).
- While the quantum Hessian estimation subroutine seems to achieve an improvement of $d$ over classical algorithms, the achieved speedups for minimax optimization in terms of quantum queries (Theorem 4.2, 4.4) are quite limited.
- The quantum advantage is formulated using query complexity. However, in Algorithms 3 and 4, it is still required to find an inexact cubic step using classical methods (e.g., solving linear systems). It is unclear if this algorithm achieves meaningful end-to-end quantum speedups.
- Besides, there are several obvious typos/formatting issues in this submission, see the questions below.

---

> ### Author Rebuttal · Authors · 2025-07-29
>
> Thanks for your helpful comments.
>
> ## To weakness 1:
>
> Our algorithm for estimating the Hessian-vector and full Hessian is an application of Jordan’s gradient estimation. However, we kindly disagree that this impacts the contribution of this work.
>
> There are no analytical results for estimating the Hessian by quantum function value oracles before. Our work is the first to study the query complexity for estimating the Hessian with an arbitrary desired accuracy $\epsilon$. We believe these oracles perform important roles in quantum optimization.
>
> In addition, our contribution is not limited to the construction of quantum Hessian evaluation oracles, but also lies on how to efficiently use such oracles to demonstrate over classical algorithms and the existing quantum algorithms (please refer to our response to limitation 1).
>
> ## To weakness 2:
>
> The state-of-the-art classical algorithm requires  $\mathcal{O}(d\epsilon^{-1})$  queries to find an $\epsilon$-saddle point, exhibiting a complexity that is linear in both $d$ and $\epsilon^{-1}$. In contrast, our algorithm (HAQZO+, Algorithm 4) can solve this problem with a query complexity that is **sublinear** in both $d$ and $\epsilon^{-1}$.
>
> A crucial contribution of our work is that the quantum speedup is achieved with respect to **both** $d$ and $\epsilon^{-1}$. This marks a significant advance, as previous quantum works in this area typically only provide speedups in the dimension $d$. We therefore believe our work offers a key conceptual contribution beyond the concrete improvement in query complexity.
>
> ## To weakness 3 and question 1
>
> We consider the case where the query complexity dominates the computation complexity of the algorithm, which is a natural setting in both classical and quantum optimization. For example, considering the generalized linear model such that  $f(x)=h(A^{\top}x)$ where $A\in R^{d\times n}$, the complexity of one query is at least $nd$, which can be much larger than the gate complexity if $n\gg d$. Our algorithm achieves meaningful quantum speedups under such a setting, typically when the dimension is not too large.
>
> The inexact cubic step is indeed computed classically. This is permissible as any classical computation can, in principle, be performed on a quantum computer. The quantum speedup we present is **not negated** by this classical step. The primary computational bottleneck in many applications is the function value query, especially when the function evaluation is handled by a complex subroutine. In such scenarios, query complexity serves as a crucial proxy for overall computational efficiency.
>
> **Furthermore, we respectfully disagree with the assertion that the presence of classical computation steps precludes a "genuine" quantum speedup.** For example, fundamental arithmetic operations, such as addition and multiplication, do not have quantum speedups. These operations are essentially performed classically within quantum algorithms. However, invoking classical arithmetic operations in quantum algorithms is not recognized as weakening the quantum speedup in general. Our position is that quantum advantage can be assessed based on the query complexity for the specific problem at hand. The optimization of the computational aspects, separate from the queries, falls within the domain of quantum compilation and quantum circuit design.
>
> ## To question 2:
>
> We first respectfully point out that the state-of-the-art classical algorithm takes $\mathcal{O}(d\epsilon^{-1})$ query complexity to find the $\epsilon$-saddle point, **which is linear in both $d$ and $\epsilon^{-1}$**.
>
> Our results demonstrate a query complexity of $\mathcal{O}(d+d^{2/3}\epsilon^{-2/3})$, which improve the dependency on both $d$ and $\epsilon^{-1}$ from **linear** to **sublinear.** This means that **our algorithm always has a quantum advantage over classical algorithms.**
>
> It is a very interesting future work to show larger quantum speedups, and we believe that the quantum Hessian oracle given in this work is important to achieve this.
>
> ## To question 3:
>
> The Gap() function is defined in Definition 1 (line 104-lin 107), we will emphasize this in the revision.
>
> ## To minor typos/formatting issues:
>
> Thanks for your careful review, we will modify them in the revision.
>
> The $f^{\*}$ defined in line 244 should be modified to $f\^*\triangleq \min\_{z}f(z)$.
>
> ## To limitation 1:
>
> We kindly argue that our technical contribution is limited and would like to emphasize the parts that the reviewer may have overlooked.
>
> Firstly, our algorithms for solving minimax problems are novel. **Directly integrating the quantum estimation into the classical algorithms leads to additional errors from the inexact gradient/Hessian oracles and the extra-gradient step.** We overcome this issue by adjusting the regularization term $\lambda_t$ and carefully setting the quantum estimation error. To the best-of-our-knowledge, even under classical settings, there are no results that allow the inexactness in both gradient and Hessian to solve the minimax problems.
>
> Secondly, the quantum Hessian estimation is a natural generalization of Jordan’s gradient estimation method and offers important oracles in quantum optimization. It can not only provide quantum speedup over classical algorithms, but also further accelerates the quantum algorithms. Fot example, our proposed QCNM method demonstrated a further acceleration upon the state-of-the-art quantum algorithms when $d\leq \epsilon^{-1/4}$.
>
>
>
> ## To limitation 2:
>
> Please refer to our response to weakness 2 and question 2.

---

> > ### Author Response · Authors · 2025-08-04
> >
> > Dear Reviewer iyXU,
> >
> > In our rebuttal, we responded to your all questions.
> >
> > We would like to know if our response has addressed your issues?
> >
> > Best regards,
> >
> > Authors

---

> ### Comment · Reviewer_iyXU · 2025-08-05
>
> I thank the authors for their kind explanation, which has addressed some of my concerns. Based on their rebuttal and discussions with other reviewers, I have updated my score. My additional comments are as follows:
>
> 1. If I understand correctly, from the discussion between the authors and Reviewer mnz2, one can obtain a quantum zeroth-order algorithm with query complexity $d + d^{2/3}\epsilon^{-2/3}$, which is comparable with the algorithm presented in this work (using quantum Hessian estimation). This is certainly an interesting and arguably stronger result. However, it also suggests that the role of quantum Hessian estimation is diminished and appears more like a corollary of Jordan's algorithm.
>
> 2. That said, I still stand by my original assessment that the quantum speedup in this paper is rather marginal. The proposed quantum algorithm requires fault-tolerant quantum hardware, and in practice, the sub-quadratic speedup presented here is unlikely to be realized due to the overhead of quantum error correction.

---

> ### Author Response · Authors · 2025-08-05
>
> Thanks for your reply and updating your score. We would like to add the following clarification on your comments.
>
> 1. In our discussion with Reveiwer mnz2, we focus on if the quantum Hessian oracle should be presented explicitly instead of obtaining a new algorithm.  In fact, if we write the quantum Hessian estimator in Algorithm 4 by calling Algorithm 2 and 3 explicitly, then we can obtain a algorithm with $\mathcal{O}(d+d^{2/3}\epsilon^{-2/3})$ with only the estimation of gradient. This is not a new algorithm, but only a different presentation of the algorirthm in this work. We kindly argue that the quantum Hessian estimation should be constructed explicitly in our response to Reviewer mnz2 for the following two reasons.
>
> * The construction of the quantum Hessian estimation allows us to bound the fail probability easily so that one can obtain a high probablity bound directly.
>
> * In classical optimization work, it is very popular to construct a Hessian estimation explicitly by gradient or function value oracles and use such an estimation to simplify their algorithms. Hence, we also think it is also proper to construct a quantum Hessian estimation explicitly.
>
> 2. In our minimax optimization, we obtain a sub-quadratic speedup, but on both the parameter $d$ and $\epsilon^{-1}$, which we believe is not marginal. For the non-convex minimization problem, our algorithm exhibits a complexity of $\mathcal{O}(d+d^{1/2}\epsilon^{-3/2})$, which is at least a quadratic speedup on the dependency of dimension upon the prior zeroth-order cubic-regularized-Newton methods $\mathcal{O}(d^2+d^{3/2}\epsilon^{-3/2})$. As compared to the prior quantum algorthm, such a speedup comes from ultilizing the quantum estimation of Hessian, which again motivates us to construct such an oracle explicitly.

---

### Official Review · Reviewer_mnz2 · 2025-07-02

**Clarity:** 4
**Significance:** 2
**Originality:** 2
**Rating:** 4
**Confidence:** 5

**Summary:**

This paper studies dimension-dependent quantum algorithms for convex-concave minimax optimization, assuming access to a quantum function value oracle. Classically, the problem can be solved using $O(d/\epsilon)$ queries to a classical oracle. In contrast, the proposed quantum algorithm achieves a query complexity of $O(d + d^{2/3}\epsilon^{-2/3})$, yielding an improvement of order $d^{1/3}\epsilon^{-1/3}$. Additionally, for the strongly convex--strongly concave case, the authors present an algorithm with a query complexity improvement of order $d^{1/3}\mu^{-1/3}$. The techniques are further extended to the non-convex setting, where the objective is to find an $\epsilon$-stationary point.

Technically, this paper introduces quantum algorithms for estimating the Hessian-vector product and the Hessian of the objective function. These techniques are used as subroutines in the main algorithm, which maintains an estimate of the Hessian and performs approximate Newton updates. In particular, the Hessian estimate is updated using a lazy updating framework. The algorithm for finding stationary points of a non-convex function follows a similar high-level approach.

**Questions:**

1. For every quantum algorithm in this paper, can we have a corresponding classical algorithm with first-order input that achieves the same query complexity, using basically the same algorithms?
2. Can we have dimension-independent quantum algorithms using an evaluation oracle? If so, can we design algorithms whose complexity interpolate between this dimension-independent bound and the bounds obtained in this paper?
3. For the dimension dependent non-convex optimization setting, there is a more recent paper "Improved Complexity for Smooth Nonconvex Optimization: A Two-Level Online Learning Approach with Quasi-Newton Methods" that might be worth discussing.

**Ethical Concerns:**

["NO or VERY MINOR ethics concerns only"]

**Final Justification:**

The authors have addressed my concerns so I updated my score.

**Limitations:**

yes

**Quality:**

3

**Strengths And Weaknesses:**

Strengths:
1. This paper is the first to discuss quantum speedups for convex-concave minimax optimization problem with function value access.
2. The paper is well-written, and the proofs are presented clearly.
3. The analysis of the lazy Hessian algorithm (Algorithm 4) is interesting and may be of independent interest

Weaknesses:
1. The query complexity bounds are not written in a very precise way, it should be $d+\cdots$, since at the beginning of the algorithm you need $d$ queries to compute the full Hessian. Omitting it is inaccurate.
2. The algorithms for estimating the Hessian-vector product and the full Hessian appear to be, to some extent, relatively straightforward applications of Jordan’s gradient estimation algorithm.

---

> ### Author Rebuttal · Authors · 2025-07-29
>
> Thanks for your helpful comments.
>
> ## To weakness 1:
>
> Thanks for pointing out this, we will update the query complexity of HAQZO+ (Algorithm 4), Restart-HAQZO+, and QCNM to $\mathcal{O}(d+d^{2/3}\epsilon^{-2/3})$, $\mathcal{O}(d+d^{2/3}(L_2/\mu)^{2/3})$, and $\mathcal{O}(d+d^{1/2}\epsilon^{-3/2})$ respectively.  The $\mathcal{O}(d)$ factor is only an addition term and can be omitted when $d$ is small ($d\leq \epsilon^{-2}$ for convex-concave problem and $d\leq \epsilon^{-3}$ for non-convex problem).
>
> ## To weakness 2:
>
> We agree that our algorithm for estimating the Hessian-vector and full Hessian is an application of Jordan’s gradient estimation. However, we kindly disagree that this impacts the contribution of this work.
>
> There are no analytical results for estimating the Hessian by quantum function value oracles before. Our work is the first to study the query complexity for estimating the Hessian with an arbitrary desired accuracy $\epsilon$. We believe these oracles perform important roles in quantum optimization.
>
> In addition, our contribution is not limited to the construction of quantum Hessian evaluation oracles, but also lies on how to efficiently use such oracles to demonstrate over classical algorithms and the existing quantum algorithms (please refer to our answer to question 1).
>
> ## To question 1:
>
> For the minimization problem, the complexity of QCNM aligns with the complexity of First-order CNM in [11].
>
> For the minimax problem, **there are no corresponding classical algorithms with first-order input that achieve the same query complexity**. Unlike FO-CRN [11] for non-convex optimization, our algorithms (HAQZO, HAQZO+) conduct an extra-gradient step, which incurs additional error from the inexact gradient. This require both novel algorithm design, i.e, the choice of the regularziation term $\lambda_t$, and careful analysis to control the error from the gradient and Hessian estimator. To the best-of-our-knowledge, such Hessian-aware algorithms with zeroth-order or first-order oracles have not been considered under classical settings before.
>
> ## To question 2:
>
> There are no dimension-independent quantum algorithms for minimax problems with a function value oracle. It is possible to design a dimention-free quantum algorithm for minimax problems by using an evaluation oracle on extra-gradient method or optimistic gradient descent ascent method. It is also a very interesting future work to design algorithms with complexity interpolate between dimension-free algorithm and HAQZO/HAQZO+.
>
> ## To question 3:
>
> Thanks for pointing out the paper. We did not discuss this paper in the current version because we focus on the results for both classical and quantum algorithms with zeroth-order oracles (in Table 3), while this recent paper uses the first-order oracle. We will discuss it in the revision.

---

> > ### Comment · Reviewer_mnz2 · 2025-08-01
> >
> > I want to thank the authors for a very detailed rebuttal. I have a follow up question on question 1: I'm following that there are no prior works giving classical algorithms with first-order input that achieve the same query complexity. However, I'm wondering if you can replace the quantumGradient and quantumHessian step in Algorithm 3 with gradient query and Hessian computation by taking the difference of d pairs of gradient queries, can you obtain a first-order classical algorithm whose query complexity is the same as the query complexity of Algorithm 3?

---

> > > ### Author Response · Authors · 2025-08-01
> > >
> > > Thanks for your follow-up question.
> > >
> > > We think it is possible to replace the quantumGradient and quantumHessian with gradient and Hessian estimation by taking the difference of $d$ pairs of gradient queries in Algorithm 3 can achieve the same query complexity (\# gradient oracles) as in Theorem 4.2 (\# function value oracles).  In such case the gradient has to be **exact** and the Hessian is **inexact**, so that one can apply the framework in [27] by letting the Hessian estimation close enough to the exact ones with $\mathcal{O}(d)$ queries of gradient.
> > >
> > > However, there are no classical results that allow both the inexactness of gradient and Hessian for minimax optimization. In other words, if one replace the quantumGradient and quantumHessian by some inexact gradient (with even small estimation error) and inexact Hessian by taking the difference of $d$ pairs of inexact gradient queries in prior algorithms [27], they may fail to converge or with worse query complexities. **Algorithm 3 overcome such issues by providing a more general framework that allows both inexactness of gradient and Hessian as compared to the prior works.** It adjusts the regularization term $\lambda_t$ by invoking the error led by gradient and Hessian ($\epsilon_g$ and $\epsilon_{H}$) and carefully sets the error levels to achieve a desired iteration complexity.
> > >
> > > Furthermore, there are no prior works which consider the case of applying a inexact lazy Hessian (Algorithm 4) for minimax problems even when the exact gradient can be accessed. In such a case, one require to control the error led from both inexactness of Hessian and the difference between Hessian at the current iteration and the one at the snapshot point. **Algorithm 4 generalizes the strategies we developed in Algorithm 3, which is the first one to analysis the inexact lazy Hessian in minimax problems while allows the inexactness of gradient.** To the best of our knowledge, such results have not been developed even under classical settings before.
> > >
> > > We hope we have addressed your concern and are happy to answer any follow-up question.

---

> > > > ### Comment · Reviewer_mnz2 · 2025-08-02
> > > >
> > > > I want to thank the authors for their quick response. I would like to clarify that I fully acknowledge the technical contribution of their classical lazy Hessian method for minimax optimization, which I find novel, interesting, and potentially useful in broader algorithmic contexts.
> > > >
> > > > My main concern is about how the results are presented. In particular, if the same query complexities of both Algorithm 3 and Algorithm 4 can be achieved by a classical first-order method, then it seems the quantum speedup comes from combining a novel classical first-order algorithm with quantum gradient estimation. In this case, the quantum Hessian estimation appears more like a corollary of the quantum gradient estimation method, rather than a central technical contribution of the work.
> > > >
> > > > Regarding the inexactness in gradient and Hessian information, I fully acknowledge the authors’ contribution in incorporating these into the analysis—it is solid technical work. However, since the focus of the paper is on query complexity, I'm not sure how much these analysis strengthens the main claim, as quantum gradient estimation can have exponentially small error using only 1 query. Subsequently, the error in the Hessian can also be exponentially small.

---

> > > > > ### Author Response · Authors · 2025-08-03
> > > > >
> > > > > Thanks for your follow-up question and your recoginition in our technical contribution.
> > > > >
> > > > > First of all, we respectfully disagree that the quantum speedup comes from combining a novel classical first-order algorithm with quantum gradient estimation. **For the classical first-order method, the lower bound of query complexity is $\Omega(\epsilon^{-1})$, simply replacing the classical gradient estimator by its quantum counterpart cannot achieve a better dependency than $\mathcal{O}(\epsilon^{-1})$**. Our accelerated query complexity of $\mathcal{O}(d+d^{2/3}\epsilon^{-2/3})$ comes from two parts, the first-part is to consider a Hessian-aware algorithm which allows us to reduce the dependency on $\epsilon^{-1}$ by a second-order update framework and the second-part is to reduce the query complexity on $d$ by taking advantage of quantum gradient and Hessian estimation. From this point of view, we think that the quantum Hessian estimation should not be regarded as corollary of gradient estimator, but a necessary component to achieve our Hessian-aware algorithms with desired query complexity.
> > > > >
> > > > > We then illustrate the difficulty of involving both the inexactness of gradient and Hessian in our algorithm framework. Our algorithm does not involve a cubic regularized Newton step, but also an extra gradient step. The the inexact gradient in the extra gradient step lead to $\frac{1}{\lambda_t}\langle v_t,z_{t+1/2}-z\rangle\leq *$  (line 422, Eq. (15)) while we aim to upper bound $\frac{1}{\lambda_t}\langle F(z_{t+1/2}),z_{t+1/2}-z\rangle$, although the $v_t$ can approximate $F(z_{t+1/2})$ with  exponentially small error, the $z_{t+1/2}-z$ can be exponetially large since $z\in R^{d}$. We overcome this issue by carefully design the regularization term $\lambda_t$ and think these analysis techniques are essential to obtain the desired query complexity.
> > > > >
> > > > > We sincerely appreicate the reviewer for the actively participation in the discussion period and hope the above response can address your concern.

---

> > > > > > ### Comment · Reviewer_mnz2 · 2025-08-03
> > > > > >
> > > > > > Thank you for the quick reply. To make sure we are on the same page, I'll write out the argument I have in mind more explicitly. Here's a list of facts I used:
> > > > > >
> > > > > > Fact 1: Given an arbitrarily precise classical gradient oracle, one can compute the Hessian to arbitrary precision using 2d queries.
> > > > > >
> > > > > > Fact 2: Given an arbitrarily precise quantum function value oracle, one can compute the gradient to arbitrary precision using 1 query (even smaller than 1/exp, anything you wish).
> > > > > >
> > > > > > Given these two facts, I think I should clarify what I meant by “classical first-order method” in the previous message. Specifically, I am referring to potentially dimension-dependent classical algorithms that use gradient queries for an objective function with Lipschitz-continuous Hessian—i.e., the setting considered in Algorithm 4. In this regime, I believe the dimension-independent lower bound $\Omega(1/\epsilon)$ is no longer applicable.
> > > > > >
> > > > > > My previous argument also relies on the following observation towards Algorithm 4, please correct me if wrong:
> > > > > >
> > > > > > Observation 1: If you replace QuantumGradient in line 7 and 11 with an arbitrarily precise gradient query, and replace QuantumHessian with an arbitrarily precise Hessian query, Algorithm 4 with the same set of parameters would still finds an eps-saddle.
> > > > > >
> > > > > > Using the facts and this observation, here're my arguments:
> > > > > >
> > > > > > Claim 1: You can have a classical algorithm that uses d+d^{2/3}/eps^{2/3} queries to an arbitrarily precise gradient oracle that finds an eps-saddle whp. This claim is obtained by combining Fact 1 and Observation 1. (On a side note, I think such a classical algorithm would be a solid technical contribution of this work if presented explicitly)
> > > > > >
> > > > > > Claim 2: Combining this algorithm with Fact 2, we directly get a quantum zeroth-order algorithm using d+d^{2/3}/eps^{2/3} queries, without incurring the quantum Hessian estimation subroutine explicitly.
> > > > > >
> > > > > > More than happy to further clarify on any of the parts above!

---

> ### Author Response · Authors · 2025-08-04
>
> Thanks for your clarification and your recoginition in our technical contribution. The facts and observation in your post are absolutely correct and we are very happy to present the classical gradient algorithm explicitly in the revision.
>
> W’d like to add some additional comments on your claims.
>
> - For the claim 1,  Lemma 4.3 studies a general update framework (8) with inexact gradient and inexact Hessian which do not differentiate the quantum or classical queries. This allows us to derive a classical upper bound of $\mathcal{O}(d+d^{2/3}\epsilon^{-2/3})$ gradient queries quickly.
> - For the claim 2, one major difference between the classical and quantum algorithm is that the measurement on the quantum estimation oracle leads to a high probablity bound  in quantum algorithms instead of a deterministic bound in classical algorithms. We need to upper bound the fail probability at every quantum estimation of the gradient and Hessian oracles to fulfill the desired probablity bound. This motivates us to present a quantum estimation oracle for the Hessian, so that one can control the fail probability of accessing a Hessian directly.  Take Algorithm 4 for example, we set the parameters in quantum estimation oracles for the Hessian, gradient at the Newton step, and gradient at the extra-gradient step as $\delta/3T$ in line 4, 7, 11 respectively, so that the fail probability of the Algorithm 4 can be bounded by $T\*3\*\delta/3T=\delta$.
> - In addition, it is common to conduct such a Hessian-estimation oracle eplicitly in classical optimization methods. For example, [Lemma 3, A] and [Lemma 3.1, 11] present a first-order (gradient) and zeroth-order implementation of the Hessian explicitly and estimate the approximation error. Hence, we think it is natural and proper to also present such a quantum estimation of Hessian explicitly.
>
> [A]. A cubic regularization of Newton’s method with finite difference Hessian approximations. Numerical Algorithms, 2022

---

> > ### Comment · Reviewer_mnz2 · 2025-08-06
> >
> > I would like to thank the authors for their detailed and prompt response. All my questions are now addressed. I understand the authors’ argument that the quantum estimation oracle leads to a high-probability bound, which differs from the deterministic bound in classical algorithms. Although there are ways this error probability can be handled without directly incurring the Hessian, I feel the current presentation is likely the most succinct from a technical perspective.
> >
> > From my perspective, I would have a more positive evaluation of the submission if the authors are willing to emphasize slightly less that the quantum Hessian estimator is one of the main technical contributions of the paper. As it is currently presented, I, as a reviewer, am expected to evaluate the technical novelty of the contributions highlighted by the authors, and I feel the novelty of the quantum Hessian estimator is more limited compared to that of the Hessian-aware algorithm. If the authors instead place more emphasis on their contribution to the Hessian-aware algorithm, the novelty, as assessed from a reviewer’s perspective, would increase.

---

> > > ### Author Response · Authors · 2025-08-06
> > >
> > > Dear reviewer mnz2,
> > >
> > > We are happy to address all of your questions. We also appreciate your suggestion on how to present the paper and your active participation in the discussion period. We will be more than glad to involve your suggestion in our revision, including giving more emphasize on our Hessian-aware algorithm rather than the quantum Hessian estimator. We believe this can be done within a minor revision.
> > >
> > > We  sincerely hope that you may reconsider your score and overall evaluation on this paper.
> > >
> > > Best regards,
> > > The authors

---

> > > > ### Comment · Reviewer_mnz2 · 2025-08-08
> > > >
> > > > Thank you again for the detailed responses, my concerns are addressed, so I raised my score.

---

> > > > > ### Author Response · Authors · 2025-08-08
> > > > >
> > > > > Thanks for your support to our work and your insightful suggestion.

---

### Note · Authors · 2025-08-12

Dear reviewers, ACs, and SACs,

We sincerely appreciate all the reviewers for their helpful comments and suggestions during the discussion period. We summarize the following points in the final remarks.

- All reviewers agreed that we have addressed their concerns in the rebuttal. We emphasize the novelty and technical contribution which may have been overlooked at the initial stage (from reviewers iyXU and 77Pf). We illusrate the reason of focusing on query complexity instead of the gate complexity (from reviewers iyXU and 77Pf) and the discuss on the lower bound (from reviewers 77Pf and iZ6k). We have also addressed reviewer mnz2’s concern on the construction of quantum Hessian estimation oracle through a multi-round discussions.
- We noticed that reviewer iyXU raised some comments based on our discussion with reviewer mnz2. We have added follow-up response to address the remaining concern. The reason to construct a quantum Hessian estimator raised by reviewer mnz2 have already been addressed. The remaing concern is that the sub-quadratic quantum speedup may be marginal, while we have achieved quadratic quantum speedup for non-convex optimization on $d$. In addition, we remark that demonstrating quantum speedup for convex-concave problems on both $d$ and $\epsilon^{-1}$ is crucial, since previous quantum works  typically only provide speedups in the dimension $d$.
- Most reviewers found our technical contribution is solid. Reviewer mnz2 comments that "it is solid technical work" and found the analysis of Algorithm 4 may of independent interest. Reviewer iyXU finds that theoretical analysis appears complete and solid. Reviewer eEJb also comments that "the paper is very technically solid" and Reviewer iZk6 finds "this is a technically solid work providing quantum algorithms with provably better performance for minimax optimization over best-known classical counterparts".

We hope take these points can be taken into the final consideration.



Best regards,

The authors

---

### Decision · Program_Chairs · 2025-09-17

**Decision:**

Accept (poster)

**Comment:**

This paper studied quantum algorithms for minimax optimization. In particular, this paper gave a quantum zeroth-order algorithm that can find an eps-saddle point within ~O(d+d^{2/3}eps^{-2/3}) function value oracle calls, improving over the O(d*eps^{-1}) classical upper bound of classical zeroth-order methods. A similar result was obtained when strong convexity and concavity present.

Reviewers found the technical results interesting. Nevertheless, some concerns are raised, in particular the time complexity of the algorithm, contribution on the quantum Hessian estimator, and other minor points. There were adequate discussions during the rebuttal, and the reviewers found the answers from authors convincing. In all, considering the overall contribution and positive scores, the decision is to accept this paper in NeurIPS 2025.

Nevertheless, for the final version of the paper, the authors should consider implementing the following changes from reviewer-author discussions as well as some suggestions from AC:
- In the complexity bounds, add the extra linear term in d where appropriate, as pointed out by reviewer mnz2.
- The authors should polish the storyline following the discussion with Reviewer mnz2. In particular, it is more appropriate to tune down quantum Hessian estimation and emphasize more on the analysis of involving both the inexactness of gradient and Hessian.
- The authors should add more discussion about the gate complexity of the proposed quantum algorithm and when it achieves a genuine quantum speedup. A starting point can be the paragraph "We consider the case where the query complexity dominates the computation complexity of the algorithm, which is a natural setting in both classical and quantum optimization… Our algorithm achieves meaningful quantum speedups under such a setting, typically when the dimension is not too large" in rebuttal.
- Regarding quantum Hessian estimation, the authors should also cite the [ZS24] paper mentioned by Reviewer 77Pf, titled "Quantum spectral method for gradient and Hessian estimation", and make a comparison with their bounds and techniques.
- Other minor points promised in rebuttals.